# The Growth-Inhibitory Effect of Increased Planting Density Can Be Reduced by Abscisic Acid-Degrading Bacteria

**DOI:** 10.3390/biom13111668

**Published:** 2023-11-19

**Authors:** Lidiya Vysotskaya, Elena Martynenko, Alena Ryabova, Ludmila Kuzmina, Sergey Starikov, Sergey Chetverikov, Elvina Gaffarova, Guzel Kudoyarova

**Affiliations:** Ufa Institute of Biology, Ufa Federal Research Centre, Russian Academy of Sciences, Prospekt Oktyabrya, 69, 450054 Ufa, Russia; evmart08@mail.ru (E.M.); alena.ryab2013@yandex.ru (A.R.); ljkuz@anrb.ru (L.K.); senik0406@gmail.com (S.S.); che-kov@mail.ru (S.C.); gaffarova2002@list.ru (E.G.)

**Keywords:** *Lactuca sativa* L., growth, planting density, plant competition, plant growth-promoting rhizobacteria (PGPR), abscisic acid

## Abstract

High-density planting can increase crop productivity per unit area of cultivated land. However, the application of this technology is limited by the inhibition of plant growth in the presence of neighbors, which is not only due to their competition for resources but is also caused by growth regulators. Specifically, the abscisic acid (ABA) accumulated in plants under increased density of planting has been shown to inhibit their growth. The goal of the present study was to test the hypothesis that bacteria capable of degrading ABA can reduce the growth inhibitory effect of competition among plants by reducing concentration of this hormone in plants and their environment. Lettuce plants were grown both individually and three per pot; the rhizosphere was inoculated with a strain of *Pseudomonas plecoglossicida* 2.4-D capable of degrading ABA. Plant growth was recorded in parallel with immunoassaying ABA concentration in the pots and plants. The presence of neighbors indeed inhibited the growth of non-inoculated lettuce plants. Bacterial inoculation positively affected the growth of grouped plants, reducing the negative effects of competition. The bacteria-induced increase in the mass of competing plants was greater than that in the single ones. ABA concentration was increased by the presence of neighbors both in soil and plant shoots associated with the inhibition of plant growth, but accumulation of this hormone as well as inhibition of the growth of grouped plants was prevented by bacteria. The results confirm the role of ABA in the response of plants to the presence of competitors as well as the possibility of reducing the negative effect of competition on plant productivity with the help of bacteria capable of degrading this hormone.

## 1. Introduction

The presence of competing weeds inevitably decreases the growth and productivity of crop plants [1,2]. Although modern agricultural practice discourages weeds and lessens their negative impact, competition between the crop plants themselves also inhibits the growth of individual plants when closely spaced. High planting densities are increasingly typical of modern broadacre farming [3,4,5,6]. The growth inhibition is explained not simply by competition between closely-spaced plants for resources in the soil, because it occurs even before the presence of near neighbors reduces availability of water and mineral nutrients [1,7]. This is a consequence of each plant’s reaction to environmental signals created by the close presence of neighbors [8,9]. 

Such signals may comprise the production of the hormone abscisic acid (ABA), leading to rapid closure of stomata [10]. This is one of the first reactions to the presence of neighbors that can inhibit plant growth [11]. Evidence for this comes from the absence of this stomatal response in ABA-deficient tomato mutant [11]. It has also been found that ABA concentrations increase rapidly in the shoots of lettuce plants in the presence of neighbors [12]. These early reactions herald future competition for basic resources such as minerals, water, and light. The latter not only involves the shading of photosynthetic radiation but also changes in morphogenetically-active radiation, such as the relative proportion of red to far red light [8]. It is known that ABA is involved in the reaction to light signals [13,14,15]. 

Significant amounts of ABA are constantly introduced into soil by roots. ABA transporters located in root epidermal cells can support this efflux of ABA [16]. Consequently, its concentration in the soil solution can gradually increase during the growing season [17]. The accumulation of ABA in soil can, in turn, influence plant resistance to water shortage since ABA accumulation during water stress maintains the plant water status by promoting stomatal closure [18]. ABA can, however, also inhibit root and shoot growth of well-watered plants [19,20]. We have previously shown [11] that increasing the planting density of tomatoes increases the pH of the soil and xylem sap, contributing to the molecular dissociation of ABA. The dissociated form is less readily taken up by cells than the undissociated form and ABA is therefore more readily transported through apoplast on its way from roots to shoots [21]. This may explain the greater accumulation of ABA in the shoots of closely competing plants compared with those spaced more widely.

The negative effects of ABA accumulation can be offset by experimentally inhibiting its synthesis using fluridone (an inhibitor of ABA synthesis). The result is a suppression of the growth-inhibiting effect of inter-plant competition [12]. These results indicate the potential for increasing plant productivity at high planting density by reducing the level of ABA in competing plants. This approach may be useful in the future since a higher seeding rate can be expected to help raise yields from the limited land area available worldwide for agriculture. Increasing yields by raising planting densities can also be beneficial for greenhouse cropping and other forms of intensive horticulture. 

Although the approach based on reducing concentration of ABA in plants at high planting densities is, in principle, not controversial, the method of achieving this goal using an ABA synthesis inhibitor has several disadvantages. Fluridone (an herbicide) inhibits not only the synthesis of ABA but also of the carotenoids [22], which are necessary for the normal functioning of photosynthetic apparatus [23]. Although in our experiments the chosen concentration of fluridone did not inhibit normal plant growth in the short term, longer exposure even at a low concentration will likely depress plant growth [12]. An alternative approach is one based on the introduction of bacteria able to degrade ABA in the rhizosphere. This ability has been demonstrated in certain bacteria (e.g., *Rhodococcus* sp. P1Y and *Novosphingobium* sp. P6W) growing on a selective medium in which ABA served as the only carbon source [24,25]. It has been shown that the introduction of such bacteria into the rhizosphere of tomato plants reduces their ABA concentration in shoots and roots [24]. 

The capacity of so-called plant growth-promoting rhizobacteria to stimulate plant growth more generally is attracting increasing attention [26,27,28]. For example, microorganisms can directly influence plant growth by synthesizing growth-stimulating hormones [29,30] and degrading growth-inhibitory hormones [31,32]. Changes in plant hormonal status may therefore result from either microbial consumption or production of hormones, or from changes in plant hormone metabolism in planta or both [33,34].

The purpose of the present work was to evaluate the ability of the *Pseudomonas plecoglossicida* 2.4-D strain to degrade ABA, as well as to evaluate its effect on ABA content in soil and within lettuce plants grown at different planting densities. This strain of bacteria is capable of growing on a medium with ABA as the only source of carbon. Thus, we tested the hypothesis that bacteria capable of degrading ABA can reduce the growth inhibitory effect of competition between plants by reducing concentration of this hormone within plants and their root environment.

## 2. Materials and Methods

### 2.1. Experimental Design

Lettuce plants (*Lactuca sativa* L., cultivar Moscow Greenhouse) were grown under controlled laboratory conditions at an irradiance of 320 μmol m^−2^ s^−1^ PAR, a 14 h photoperiod, 18–26 °C, and air humidity of 40–50%. Lettuce seeds were germinated in the soil and after the third true leaf appeared on the seedlings, they were transplanted into 0.2 L pots (either individually or in groups of three, simulating different planting densities) with 0.2 kg of sand soaked with Hoagland–Arnon solution (5 mM KNO_3_, 5 mM Ca(NO_3_)_2_, 1 mM KH_2_PO_4_, 1 mM MgSO_4_) until completely saturated. Prior to experiments, sand was sterilized by calcinations to exclude the presence of undesirable bacteria. Plants were watered each day to reach 70–80% of full water capacity, by adding to all pots the same volume of nutrient solution (100% H-A) as a top dressing and the required amount of water, depending on transpiration losses. One day after planting, one and three plants per pot, a suspension of cells of *Pseudomonas plecoglossicida* 2.4-D strain (GenBank KY593189.1) from the collection of microorganisms of the Ufa Institute of Biology [35], was introduced into the rhizosphere (to the surface around each seedling). In this case, two types of plant treatment were used: increased planting density (placing 3 plants in a pot instead of one) and rhizosphere inoculation with *Pseudomonas plecoglossicida* 2.4-D strain. Single non-inoculated plants served as a control. 

### 2.2. Bacterial Strain and Culture Media

*P. plecoglossicida* 2.4-D has been isolated from soil contaminated by waste from petrochemical production. The cells of the strain are Gram-negative moving bacteria with a diameter of 1.0 μm and a length of 2.5–3.0 μm. The optimal growth was at a temperature of 26–30 °C, and the optimal pH was 6.8–7.2. 16S rRNA gene sequence similarity allowed identification of the strain as belonging to *P. plecoglossicida* [35,36]. It was shown that the strain is capable of synthesizing and excreting indole-3-acetic acid (IAA) into the external environment [36].

To obtain the bacterial preparation, microorganisms were grown in liquid King B medium [37] in 100 mL of nutrient medium in Erlenmeyer flasks (250 mL) on an Innova 40R shaker incubator (New Brunswick, NJ, USA) at 28 °C and rotated at a speed of 160 rpm for 2 days. Bacterial cells were separated by Sigma 2–16PK centrifuge (Wiegand Int. GmbH, Hamburg, Germany) at a speed of 8000 rpm at 10 °C, and suspension of microorganisms diluted in sterile tap water was added to the pots to yield 3.8 × 10^6^ and 3.8 × 10^8^ CFU per g of soil thereafter designated as the 10^6^ and 10^8^ treatments, respectively. An aliquot of sterile tap water was added to the control pots. 

### 2.3. Growth Rates, Pigment Content and Nitrogen Index 

At 4 and 12 days after inoculation (5 and 13 after transplantation) the fresh weight of shoots and roots (measured immediately harvesting), leaf area, pigment content, and nitrogen index of lettuce plants were determined. Leaf area of lettuce plants was measured using a Li-3100 automatic area meter (LICOR, Lincoln, NE, USA). Nitrogen balance index (NBI), calculated as the ratio between chlorophyll and flavonols contents has been shown to be a very sensitive indicator of crop N status [38]. NBI and content of the total chlorophyll in leaves epidermis were measured in all formed leaves of lettuce plants using the Dualex scientific+ (Force-A, Paris, France).

### 2.4. Abscisic Acid Assay

ABA was measured in sand washes, culture liquid, and lettuce plants. At the end of experiments, 2 h after the last watering, samples of shoots and roots were fixed in liquid nitrogen and soil washes were collected to determine the hormone content in them. To obtain soil washes, water was added to the sand soil remaining after removing plants from the pot, and liquid was separated by filtration. 

#### 2.4.1. Extraction of Abscisic Acid

To determine ABA in plants, samples of shoots and roots of single and competing lettuce plants fixed with liquid nitrogen were homogenized and hormone was extracted with 70% ethanol overnight. The alcohol extract was evaporated to an aqueous residue, and after centrifugation, aliquots of the supernatant and soil washes were taken for further purification. This was carried out according to a modified scheme with a decrease in the volume of extractant [39]. After adjustment of pH to 2.5 with HCl, the extract was partitioned three times with diethyl ether (3:1 ratio of organic to aqueous phases). Subsequently, ABA was transferred from the organic phase into 1% sodium hydrocarbonate (pH 7–8) (1:3 ratios of the aqueous to organic phases). Readjusting the pH of the aqueous phase to 2.5 and re-extracting with diethyl ether gave the final extract. Reducing the amount of extractant at each stage of extraction and re-extraction increased selectivity of hormone extraction, while recovery was not less than 80%.

#### 2.4.2. Enzyme Immunoassay 

Hormones were immunoassayed using the corresponding specific antibodies. Competitive variant of enzyme-linked immunosorbent assay (ELISA) was carried out using protocol as described [40]. Protein–phytohormone conjugate was adsorbed to a 96-well polystyrene microtiter plates in phosphate buffer (pH 7.5) at 37 °C for 1.5 h. The plates were washed thrice with phosphate-buffered physiological solution (100 mM NaCl) containing 0.05% Tween 20 and 0.5% ovalbumin (pH 7.2). A mixture of 10 µL of ABA sample or different concentrations of ABA standard mixed with 180 µL of antisera against ABA was added to each well and incubated for 1 h at 37 °C. At this stage, hapten of the sample (ABA) competes with a reference antigen (ABA conjugate with protein bound to the well walls) for binding to a specific amount of an antibody. After washing out the unbound rabbit serum, goat antirabbit IgG conjugated to peroxidase was incubated with the adsorbed antigen–antibody complex for 1 h at 37 °C. After washing the wells, the substrate solution, consisting of o-phenylene-diamine (1 g L^−1^): 0.3 M phosphate buffer pH 5.8: 3% hydrogen peroxide in the ratio of 10 mL: 15 mL: 50 µL, was added. Development of color was quantified at 492 nm with a microphotometer (Uniplan, Moscow, Russia). Reliability of the method was confirmed by comparison of its results with the data obtained with HPLC combined with mass spectrometry [40] and in the present work. 

### 2.5. Measurement of the Content of Abscisic Acid in the Culture Liquid Using HPLC-MS

To study degradation of ABA by bacteria, they were grown for 10 days in Raymond’s liquid medium [41], to which ABA was added to a final concentration of 100 mg/L under conditions described in Section 2.2.

The culture liquid was subjected to centrifugation at 8000 g followed by ultrafiltration through cassettes with a pore diameter of 1 kDa (Sartocon slice cassette, Mainz, Germany).

Samples were analyzed on an LC-20 Prominence HPLC system with an SPD-M20A diode array detector (Shimadzu, Tokyo, Japan). A PerfectSil Target ODS-3 HD 5 µm (150 × 4.6 mm^2^) column (MZ-Analysentechnik, Mainz, Germany) was used. A 50% solution of acetonitrile in 0.1% acetic acid was used as the mobile phase at an elution rate of 0.4 mL/min. The volume of the injected sample was 5 μL. HPLC–MS analysis was performed on an LCMS-IT-TOF tandem liquid chromatography–mass spectrometer (Shimadzu, Tokyo, Japan) using electrospray ionization (ESI) in the negative ion mode. The trifluoroacetic acid solution was used as a standard to adjust the sensitivity and resolution, and to perform the mass number calibration. ABA concentration was calculated from a calibration curve constructed using a standard (Sigma-Aldrich, St. Louis, MI, USA).

### 2.6. Statistics

Data were processed by means of one-way analysis of variance (ANOVA) with Duncan’s multiple range tests used to discriminate means (*p* ≤ 0.05) using Statistica version 10 (Statsoft, Moscow, Russia).

## 3. Results

### 3.1. Effects of Competition and Bacterial Treatment on Soil Humidity and Plant Nitrogen Index

Over the first 2–4 days of the experiment, the water contents of pots containing a single plant or three plants, measured immediately before watering, were not statistically different from each other. It was about 65% of the total water capacity, which corresponds to normal soil water availability (Figure 1). On day 12, water content measured before watering event in the 3-plant pots was approximately 25% of the total water capacity, indicating water shortage by this time (Figure 1). The water content in the substrate of the grouped inoculated plants decreased to the level of 20% of the full water capacity of sand soil before the next watering event. At the same time, in pots with one plant, soil humidity was kept at 60%. The soil water content seemed to additionally decrease in the 10^8^ treatment (compared to both non-inoculated plants and 10^6^-inoculated plants) in both individually and densely planted plants. It is likely that the higher bacterial inoculum might have stimulated the plants to extract the water from the soil more efficiently than in the absence (or presence of lower inoculum) of bacteria.

Measurement of the nitrogen index 5 days after the start of competition experiment showed no differences in this indicator between lettuce plants grown individually or in the presence of neighbors, indicating the absence of nitrogen deficiency (Figure 2A). On the 13th day, the nitrogen index in grouped non-inoculated plants decreased by 10% compared to single plants (Figure 2B); in inoculated plants an increase in planting density did not lead to a decrease in the nitrogen index.

### 3.2. Effects of Competition and Bacterial Treatment on the Growth and Chlorophyll Content of Lettuce Plants

On the fifth day after the start of competition experiment, the fresh weight of shoots and roots of each of competing plants was lower than that of plants grown individually, (Figure 3A,B). Thus, increasing planting density inhibited plant growth. The lack of water or nitrogen deficiency found early in the competition experiment (Figure 1 and Figure 2) suggested that by this time growth inhibition in the presence of neighbors was not caused by resource scarcity. 

On the fifth day after the start of competition experiment, bacteria stimulated accumulation of shoot biomass in grouped plants (by 13 and 36% at an inoculant concentration of 10^6^ and 10^8^, respectively), but did not promote it in single plants. As a result, the weight of shoots and roots of each of the three grouped plants were not lower than that of single plants, when bacteria were introduced into rhizosphere (Figure 3A,B). Thus, the introduction of microorganisms of *P. plecoglossicida* 2.4-D strain into the rhizosphere of plants prevented the negative effect of increased planting density on plant growth.

Later on, the growth-inhibiting effect of increased planting density became more pronounced (Figure 3C,D). By this time, the presence of neighbors reduced the availability of resources (water and nitrogen) (Figure 1 and Figure 2), which might have contributed to the observed inhibition of the growth of lettuce plants with an increase in their planting density. 

The growth-stimulating effect of bacteria on shoot mass persisted 13 days after the start of experiment. (Figure 3C and Appendix A). In single plants, the weight gain caused by bacteria was 6–17%, while in competing grouped plants it was 17–52% (the higher the inoculum density, the greater the effect). The effect was again more pronounced in the case of grouped plants. At this time, the inhibitory effect of increased planting density on shoot growth was manifested in both bacteria-treated and untreated plants. However, in the first case it was less pronounced than in the second: the weight of grouped plants decreased by 35% compared to single plants in the case of non-inoculated plants, by 27%—in plants treated with bacteria at a dose of 10^6^, and by 24%—with a bacterial concentration of 10^8^. Thus, inoculation diminished the difference between the weight of single and competing plants.

Five days after the start of experiments, a tendency of competition-induced increase in leaf area was detected, which could be a compensatory response to a signal of the forthcoming shading (Figure 4A). This effect was statistically significant in the non-inoculated plants, while in the inoculated plants it was observed too, but was not confirmed as statistically significant. However, the effect disappeared later, and plant responses in terms of leaf area resembled the changes in the shoot weight (Figure 4C). By this time, leaf area was decreased by competition, but inoculation with bacteria diminished the difference between the leaf weight of single and competing plants.

Five days after the start of experiments, we detected no effect of either competition or inoculation on the root-to-shoot fresh mass ratio (Figure 4B). However, competition later (on the 13th day) increased the ratio of the mass of roots and shoots by 1.4 times in non-inoculated plants (from 0.45 to 0.62 in single and grouped plants, respectively (Figure 4D)). This effect of competition on the root-to-shoot ratio was decreased by inoculation, so the difference between single and competing plants was less pronounced in inoculated plants. Grouped plants showed a trend towards an increase in chlorophyll content compared to single plants (Figure 5A), which was probably a compensatory reaction to a signal of impending shading as planting density increased. This relationship, however, was confirmed by Duncan’s test as statistically significant only at the 10^6^ treatment. Over time (13 days after planting and bacterial treatment), this trend disappeared and no difference in chlorophyll was found between the treatments (Figure 5B). Disappearance of the difference in the pigment content excludes changes in chlorophyll content from among the possible causes of plant growth inhibition at increased planting density.

### 3.3. Effects of P. plecoglossicida 2.4-D on Concentration of Abscisic Acid in Bacterial Culture Media, Sandy Soil, and Lettuce Plants

When bacteria were cultivated in vitro on a medium containing ABA as the sole source of carbon, the concentration of bacteria increased and the content of ABA in the medium decreased from 100 to 67 mg per L (Figure 6). An immunoassay of ABA in the samples gave similar results to those of physicochemical assay (58 ± 6 mg per L (*n* = 12), at the end of incubation).

The content of ABA in the sand and shoots of plants increased with an increase in planting density, while bacterial inoculation reduced the level of ABA accumulation both in sand and in plant shoots (Figure 7).

## 4. Discussion

### 4.1. Effects of Resource Availability on the Growth of Lettuce in the Presence of Neighbors and Reduction in Growth Inhibition upon Rhizosphere Inoculation with P. plecoglossicida 2.4-D

The presence of neighbors (three plants in a pot instead of one) inhibited the accumulation of shoot and root mass as well as the leaf surface formation of the plants. Five days after the start of the competition experiment, growth retardation was not due to resource scarcity, since at this time neither the soil moisture in pots containing single and competing plants nor the nitrogen content of single and grouped plants were statistically different from each other. An increase in the mass ratio of roots and shoots (relative activation of root growth) is a classic response to water deficiency [42,43] and the absence of changes in this indicator is in accordance with the absence of water shortage detected at the beginning of the competition experiment. However, competition subsequently increased the root-to-shoot mass ratio of non-inoculated plants in accordance with the temporary decrease in soil water content in the pots with grouped plants. In contrast to the non-inoculated plants, grouped plants treated with *P. plecoglossicida* 2.4-D did not show an obvious increase in allocation to root growth, i.e., they did not exhibit the characteristic growth response to water deficiency. It appears that bacteria-treated plants did not suffer from water shortage or nitrogen deficiency and water uptake was more efficient in inoculated plants compared to single plants without investing in root growth. 

### 4.2. Involvement of Abscisic Acid in Competition Induced Inhibition of Lettuce Growth

We have previously shown [11] that inhibition of growth at increased planting density may be due to the accumulation of ABA in competing plants, since this hormone can inhibit plant growth, and the use of fluridone (ABA synthesis inhibitor) reduced the short-term growth-inhibiting effect of increased planting density [12]. In the present work, we registered accumulation of ABA in the sandy soil of pots with three plants, as well as in shoots of competing lettuce plants. It was important to test a hypothesis that introduction of bacteria capable of degrading ABA into the rhizosphere can reduce the growth inhibitory effect of competition. In the present work, we used a strain capable of growing on a medium with ABA as the only carbon source. 

#### 4.2.1. Bacterial Effects on Concentration of Abscisic Acid in Culture Media, Soil, and Plants

In the course of the present study, it has been shown that in vitro cultivation of *P. plecoglossicida* 2.4-D on a medium supplemented with ABA reduces concentration of this growth regulator in the medium. In addition, the introduction of these bacteria into the rhizosphere of lettuce plants reduced the concentration of ABA in the soil solution of pots with three plants and in the lettuce shoots growing in the presence of neighbors. The results confirm that this bacterial strain was capable of reducing the concentration of ABA not only in vitro, but also in vivo, which leads to a decrease in the concentration of ABA in plants. Our results are consistent with the data on a decrease in the concentration of ABA in plants under the influence of bacterial strain of the genus *Rhodococcus*, which was also capable of degrading ABA [24]. Thus, bacteria-induced decrease in ABA accumulation in the substrate and shoots can prevent growth inhibition in short-term competition experiment and decrease the growth inhibitory effect of prolonged presence of neighbors. These results confirm the role of ABA in the response of plants to the presence of neighbors and possibility of reducing the negative effect of competition on plant productivity with the help of bacteria capable of degrading this hormone. With prolonged exposure to *P. plecoglossicida* 2.4-D, the growth rate increased not only in grouped plants, but also in single ones. Apparently, stimulation of plant growth under the influence of these bacteria is explained not only by their ability to degrade ABA, but also by other properties (for example, the ability to produce auxins [36]). However, the effect of these bacteria on the growth of grouped plants became apparent earlier and to a greater extent than in the case of single plants. ABA concentration increased in uninoculated grouped plants, which correlated with inhibition of plant growth due to competition, whereas inoculation of plants with ABA-degrading bacteria led to a decrease in ABA concentration in grouped plants, which was accompanied by a decrease in growth inhibitory effect of competition. These results confirm the importance of the ability of bacteria to degrade ABA for the control of the growth of grouped plants that had increased concentration of ABA when not treated with bacteria. 

#### 4.2.2. Synthesis of Abscisic Acid by Plants and Its Bacterial Degradation 

With prolonged exposure to increased planting density, the accumulation of ABA in non-inoculated plants could be a response to a temporary decrease in soil moisture, which was detected just before each watering of pots with three plants. It is well known that a decrease in soil moisture activates the synthesis of ABA in the roots and increases delivery of this hormone to the shoot [44]. Although competing plants extracted more water from the pots at the end of experiment, the increased water loss was offset by daily watering, which maintained soil moisture levels sufficient to support the growth of the inoculated grouped plants. However, an increase in the level of ABA in grouped plants untreated with bacteria apparently reduced the growth rate of plants. This explains the growth retardation of non-inoculated grouped plants not so much by a decrease in soil moisture, but by the growth-inhibitory function of ABA, the accumulation of which occurred in response to a transient decrease in soil moisture in pots with grouped plants.

Plants secrete organic compounds, including hormones, through their roots [45]. The outflow of ABA is supported by transporters located in the cells of root epidermis [16] which gradually increase its concentration in the soil during the growing season [17]. Therefore, the increased concentration of ABA in the pots with grouped plants is not surprising, given that not one, but three plants released ABA into the rhizosphere. In turn, many substances are taken up by roots from the rhizosphere, including ABA present in the soil solution, resulting in an equilibrium between ABA concentrations in soil and plants [17]. This circulation of ABA out of and into plants may explain the higher concentration of ABA both in the sand solution and in the shoots of grouped plants. In turn the presence of bacteria capable of degrading ABA in the rhizosphere obviously reduces its concentration not only in the sand solution, but also in plants. A similar pattern was found in the case of bacteria capable of degrading ACC (a precursor of ethylene). The presence of these bacteria in the rhizosphere decreased ethylene production in the plants [31]. A similar explanation may be applied to our experiments where the presence of ABA-degrading bacteria in the rhizosphere was accompanied with a decline in ABA concentration in the shoots of the competing plants. The ability of bacteria to influence the ABA concentration in plants might also be due to their effects on the metabolism of this hormone in plants themselves. Effects of another bacterial strain on expression of the genes involved in the control of ABA metabolism in plants have been shown by us recently [34]. It remains a mystery how ABA accumulates in the shoots of grouped plants, while its concentration in the roots does not change. As mentioned above, the increase in planting density results in an increase in the pH of soil and xylem sap contributing to ABA dissociation and increased effectiveness of its transport from soil to roots and from roots to shoots [21]. The dissociation of ABA decreases its uptake by cells and ABA is therefore more readily transported through apoplast on its way from roots to shoots [21]. This may well explain the greater accumulation of ABA in the shoots of competing plants. Still, the distribution of ABA in plants and mechanisms regulating its transport remain an important but not fully understood process. It is hoped that further study of newly-discovered ABA transporters will help unravel this problem [44].

Although bacteria-induced decline in ABA concentrations in competing plants produced beneficial effects in the present experiments, the results of bacterial inoculation may be different under stress conditions. Our recent experiments have shown that the treatment of barley plants with another bacterial strain increased their salt resistance due to bacteria-induced accumulation of ABA in the roots, which strengthened apoplastic barriers, thereby preventing penetration of toxic ions [46]. It will be important to check the effects of ABA-degrading bacteria on plant salt resistance as well as other abiotic stresses such as heat, drought, freezing, and others.

## 5. Conclusions

The purpose of this work was to test the hypothesis that competition-induced inhibition of plant growth is caused not only by water and nutrients shortage, but also by an increased accumulation of abscisic acid, which serves as a signal indicating the presence of neighbors. This hypothesis was confirmed by obtained data. First, we showed that grouped lettuce plants experienced growth retardation in the absence of signs of water or mineral deficiency. Initially, the water content of pots with grouped plants was not different from that of single plants, while measurements of nitrogen index confirmed that grouped plants were not suffering from nitrogen deficiency at this point of time. However, ABA accumulated in the grouped plants, which could contribute to inhibiting the growth of these plants. The participation of ABA in the growth response to the presence of neighbors was confirmed in experiments on the introduction of ABA-degrading bacteria into the rhizosphere of plants. Experiments with bacterial inoculation showed a decrease in the concentration of ABA both in the soil and in plants, which was accompanied by a reduced inhibition of plant growth by the presence of neighbors. A recent publication notes that the selection of density-tolerant plants can be a promising approach to increase plant productivity and produce sustainable yields [47]. Our results suggest that an alternative approach to solving this problem may be to use bacteria that can degrade ABA and thereby reduce the growth inhibitory effect of increased planting density. However, application of this approach will depend on further study of the effect of inoculation with ABA-degrading bacteria on plant resistance against abiotic stress.

## Figures and Tables

**Figure 1 biomolecules-13-01668-f001:**
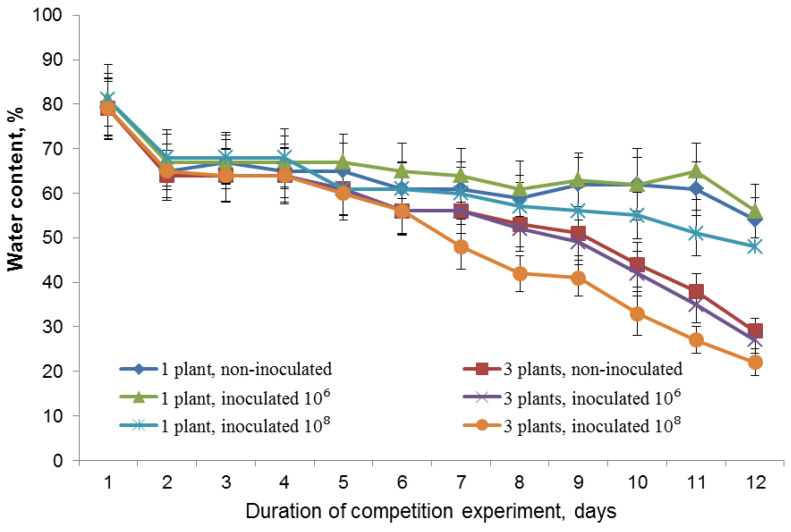
Dynamics of water content of sandy soil in pots containing individual plants or groups (three per pot), expressed as a % of total water capacity, measured immediately before watering. The rhizosphere was inoculated with bacteria (10⁶ and 10⁸) one day after planting. *n* = 12.

**Figure 2 biomolecules-13-01668-f002:**
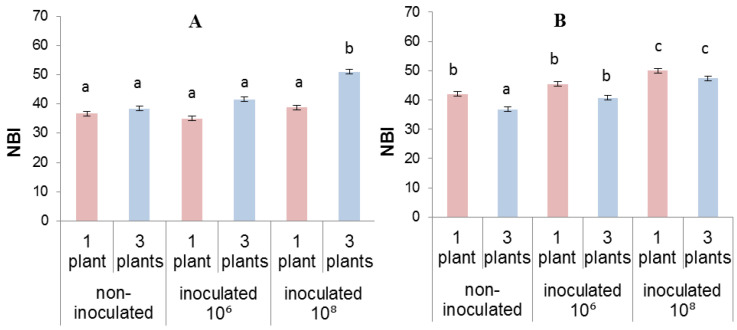
Nitrogen index (NBI) measured after 5 (**A**) and 13 (**B**) days of growing lettuce plants individually or in groups of three. The rhizosphere was inoculated with bacteria (10⁶ and 10⁸) one day after planting. Means ± SE are presented. Statistically different means (*n =* 40) are indicated by different letters (*p* < 0.05, ANOVA, Duncan’s test).

**Figure 3 biomolecules-13-01668-f003:**
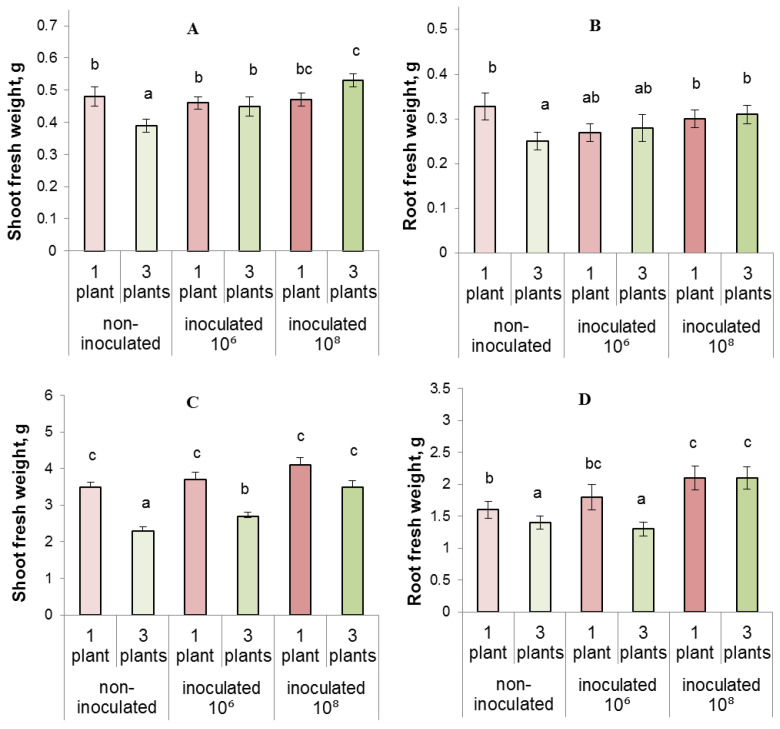
Shoot (**A**,**C**) and root (**B**,**D**) fresh weight measured after 5 (**A**,**B**) and 13 (**C**,**D**) days of growing lettuce plants individually or in groups of three. The rhizosphere was inoculated with bacteria (10⁶ and 10⁸) one day after planting. Means ± SE are presented. Statistically different means (*n* = 12) are indicated by different letters (*p* < 0.05, ANOVA, Duncan’s test).

**Figure 4 biomolecules-13-01668-f004:**
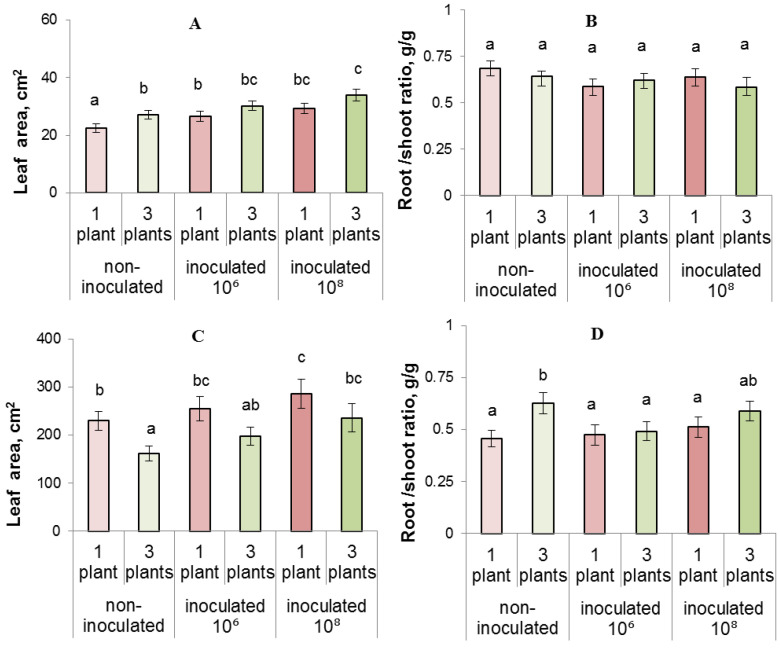
Leaf area (**A**,**C**) and root-to-shoot fresh weight ratio (**B**,**D**) measured after 5 (**A**,**B**) and 13 (**C**,**D**) days of growing lettuce plants individually or in groups of three. The rhizosphere was inoculated with bacteria (10⁶ and 10⁸) one day after planting. Statistically different means (*n* = 12) are indicated by different letters (*p* < 0.05, ANOVA, Duncan’s test).

**Figure 5 biomolecules-13-01668-f005:**
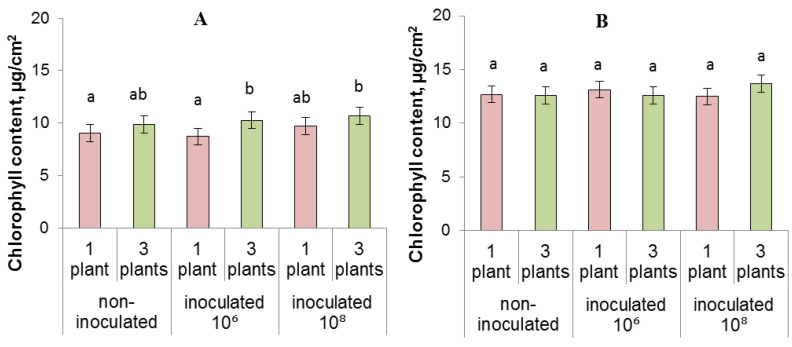
Chlorophyll content (µg/cm^2^) measured after 5 (**A**) and 13 (**B**) days of growing lettuce plants individually or in groups of three. The rhizosphere was inoculated with bacteria (10⁶ and 10⁸) one day after planting. Statistically different means (*n* = 40) are indicated by different letters (*p* < 0.05, ANOVA, Duncan’s test).

**Figure 6 biomolecules-13-01668-f006:**
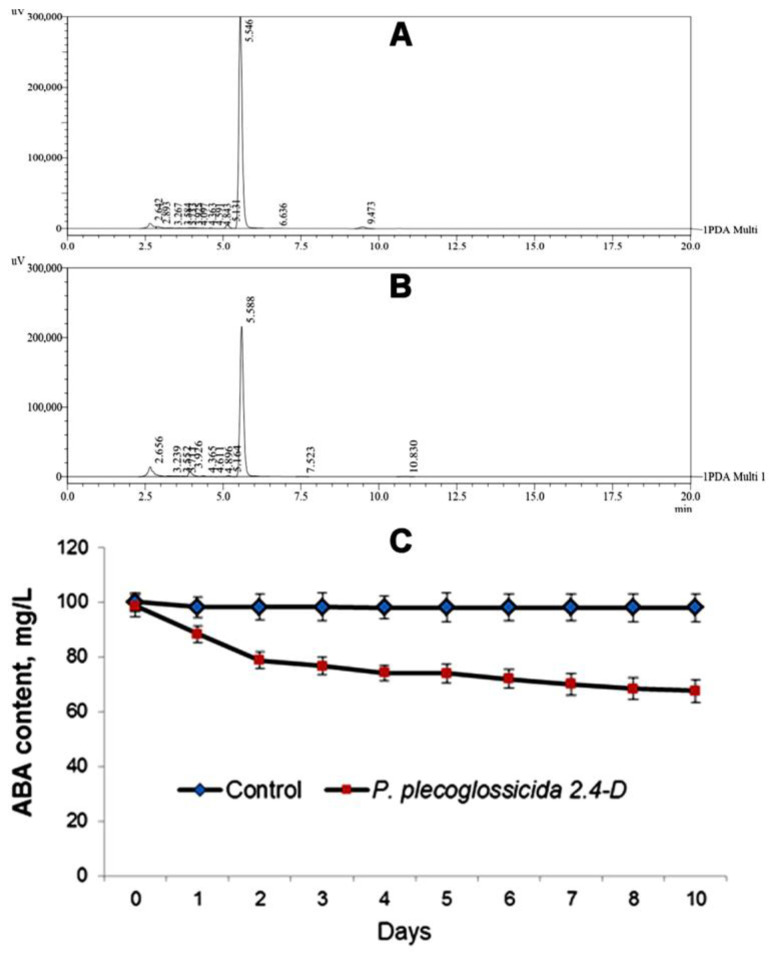
Chromatographic profiles of ABA ((5.58 ± 0.03) min release time) in the control (without bacteria) (**A**) and in the culture media of *P. plecoglossicida* 2.4-D (**B**). C—dynamics of ABA content in the control medium and in the medium with bacteria (**C**), *n* = 9.

**Figure 7 biomolecules-13-01668-f007:**
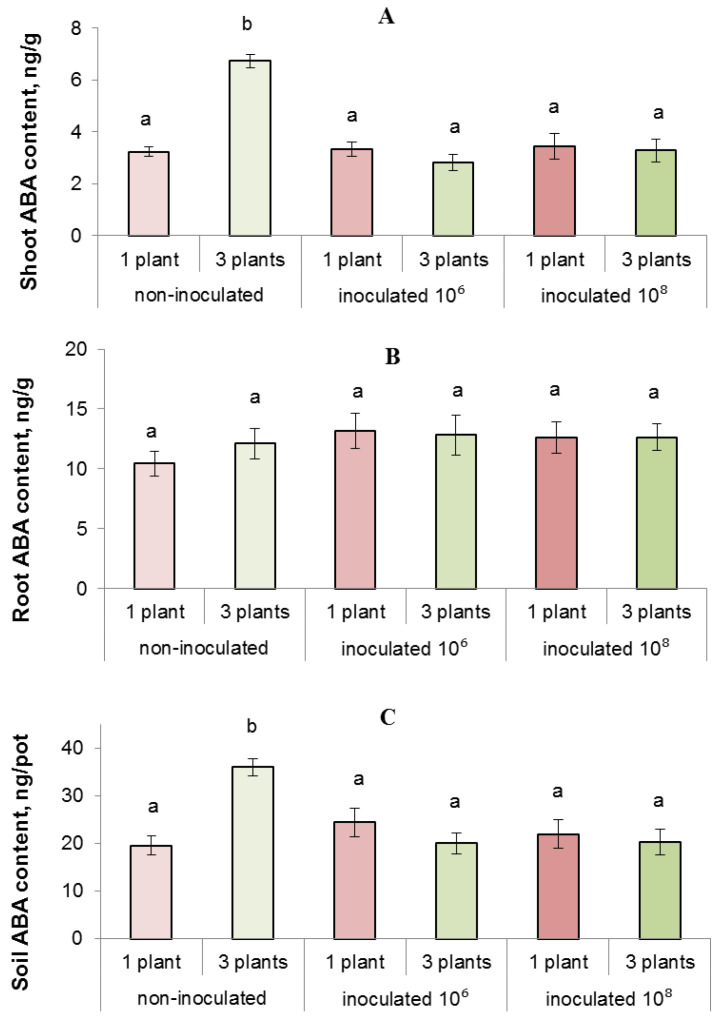
ABA content per g fresh weight of shoots (**A**) and roots (**B**) and per pot with sand soil (**C**), measured after 13 days of growing lettuce plants individually or in groups of three. The rhizosphere was inoculated with bacteria (10⁶ and 10⁸) one day after planting. Statistically different means (*n* = 12) are indicated by different letters (*p* < 0.05, ANOVA, Duncan’s test).

## Data Availability

The data presented in this study are contained within this article.

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
