# Peer review of "The Growth-Inhibitory Effect of Increased Planting Density Can Be Reduced by Abscisic Acid-Degrading Bacteria"

_biomolecules, 2023, doi:10.3390/biom13111668_

Round 1
Reviewer 1 Report
Comments and Suggestions for Authors
Dear Authors,
I have reviewed your manuscript "ABA-degrading bacteria reduce the growth-inhibitory effect of increased planting density of lettuce plants", submitted for publication in Biomolecules.
After reading your manuscript, I can tell that your results are interesting and they raise interesting and useful implications. However, the manuscript suffers from multiple weaknesses, among which I would point out a very sloppy presentation of results, a weak discussion, and scientifically inappropriate language as the most important ones. For more detailed remarks, please consult my comments as given below:
· Manuscript title - Please spell out 'abscisic acid' fully in the manuscript title ("Abscisic acid-degrading bacteria...")
· Scientific and English language - The English language of the manuscript requires moderate revision. However, the use of scientific language is very bad. I urge the Authors to perform an extensive revision of the scientific language of the article. I will give you some examples below, but I urge you to ask for revision of the scientific language of the entire manuscript by a senior scientist, preferably one who has extensive working experience from a lab abroad.
o The bacteria do not destroy abscisic acid, they degrade it. Please thoroughly revise throughout the manuscript text, using the "Find/replace" tool (lines 18, 26, 64, 156, 267, 289, 305, 309; possibly elsewhere as well)
o lines 38 and 41: Please replace "was" with "is"
o line 58: carotenoids
o lines 66-67 and 292: ABA served as the only carbon source
o line 82: either individually or in groups of three
o line 101: indole-3-acetic acid
o the lowercase letters p (for confidence interval) and n (for sample size) should always be written in italic letters. Please correct in line 174 and within all figure captions.
o line 193: What is "the action of increased plant density"? Kindly revise to meet the standards of scientifically appropriate language
o line 205: Between the treatments, not "between the options"
o Figure 4 (both the figure caption and the figure itself): square centimeter should be written as cm2, not "cm2" nor "sm2".
o line 236-237: "accumulation of shoot weight" - please revise to meet the standards of scientifically appropriate language
o line 250: Please always use English language
o line 263: There is no such thing as a "physic-chemical assay". Please check online the correct way to translate a word into English before you use it in your paper.
o line 282: The nitrogen index cannot be "the same", since the treatment is not the same. It is not even equal, let alone same. I believe that you wanted to say that they are not statistically significantly different from each other.
o line 294: Please do not call ABA "this substance". Replace it with a more appropriate term, such as "this growth regulator".
o line 325: What is "transpient"? Did you mean "transient"?
· Introduction - Apart from language imperfections, the Introduction section is virtually flawless. The problem background is correctly narrated and naturally leads the reader towards the very well-defined research goal. However, it is difficult to read because it is presented as a single block of text. Division of the introduction into separate paragraphs would be necessary to suggest to the reader what are the logical parts of the Introduction. For instance, a new paragraph is needed starting from line 55 ("Although the approach..."), when the reader gets to know why natural alternatives to ABA inhibitors are needed. Similarly, paragraph separators should be introduced elsewhere in the Introduction to suggest the logical structure of the Introduction to the reader. I recommend the division of the Introduction section into 3-4 paragraphs in total.
· Experimental design - Your experimental design is well-planned and correct. I would say that it does fulfill minimal requirements for a scientific publication. I would be happier if your experimental design was more thorough and comprehensive (please see below) because a more thorough experimental design would make the evidence for your claims more compelling and more scientifically sound. However, I still find your findings and conclusions scienifically valid and sufficient for a (modest) publication. Nevertheless, I want to share with you my thoughts about what additions I would have liked to see in your experiment, so that you might plan your experimental designs more thoroughly in your future research.
o Planting densities - I would have liked to see a broader variety of "planting densities" instead of just 1 and 3 plants per pot. If you had used 1, 2, and 3 plants per pot, or 1, 3, and 5 plants per pot in bigger pots, it would be interesting to see whether you would have observed obvious trends, or if there would be a degree of randomness to the measured data. Your results obtained with 1 and 3 plants per pot are still valid anyway, and can be considered sufficient for drawing conclusions - but more treatments always gives more information.
o Abiotic stress response - A more serious remark regarding your experimental design concerns the ABA-mediated response of plants to abiotic stress. Everyone can agree that the ABA-mediated growth inhibition in response to dense planting is counter-productive in agronomical terms in the absence of abiotic stress. However, the biological sense of this response results from plant evolution in conditions where harsh abiotic stress frequently occurred. Under abiotic stress, the ABA regulatory module is switched on, promoting a slow-down in plant metabolism and growth, in favor of energy saving and allocation of resources to the activation of tolerance mechanisms. For that reason, in stress conditions, the ABA regulatory module also downregulates other phytohormonal modules, that are related to growth and biomass accumulation in optimal environmental conditions (such as the one regulated by cytokinin, for instance, see Liu et al. 2013: https://doi.org/10.1186/1471-2164-14-594). In this light, the use of ABA-degrading microbes is likely to become counter-productive when the plants are subjected to abiotic stress, because they might negatively interfere with the adaptive plant responses to abiotic stress. Having this in mind, I would have liked to see, among your treatments, how plant growth (or survival) would be affected by the addition of Pseudomonas plecoglossicida when the plants are subjected to abiotic stress (for instance, drought, freezing, salinity, heat stress...). You might want to address this question in your next research.
· Materials and Methods - The M&M section is mostly well-written, but occasionally certain clarifications are needed:
o line 108: Please add clarification: "was added to the growing pots one day after the seedlings were transplanted, to yield...".
o line 108: There is a major issue with your article regarding the microbial density in the two bacterial treatments. In line 108, you claim that the two treatments consisted of 3.8 x 106 and 3.8 x 108 CFU per g of soil. However, in the rest of the manuscript, you refer to these treatments as "106" and "108", and the 3.8 factor is lost. This can be addressed if you corrected the names of the treatments to "3.8 x 106" and "3.8 x 108" everywhere throughout the manuscript, including all Figures, and Figure captions. This is a lot of work, and also might make the text more difficult to read. A more simple solution might be to add a clarification to the text in lines 108-109: "3.8 x 106 and 3.8 x 108 CFU per g of soil, hereafter designated as the '106' and '108' treatments, respectively".
o line 111: Please add a clarification: "the fresh weight of shoots and roots"
o line 114: Please provide a brief explanation about the meaning of the NBI index, so that the readers don't need to look for the source reference to read about it. This is very important, since in your paper, you draw important conclusions from your measurements of NBI, so it is necessary that the readers are able to readily grasp its meaning.
o line 118: Please fully spell out "Abscisic Acid" in the title of the subsection 2.4. Abbreviations should be fully spelled out in section and subsection titles even though they have been spelled out in the manuscript text previously.
o line 144: Which antisera? Please specify.
o line 166 (and possibly elsewhere as well): Please always write "L", "mL", and "µL" with a capital L.
o Section 2.6: Which software was used for statistical analysis? Please fully spell out the software name, company name, city, and country.
· Results - The Results section of your manuscript is in a very sloppy state, for which it is in sharp contrast with some other parts of your manuscript, such as Introduction, which is very tidy. The scientific language is in its poorest shape in the Results section compared to the rest of the manuscript, and overall, it looks like this section has not been revised by a senior scientist. Also, the results are displayed in chaotic order both in the Figures, and in the narration of the results within the manuscript text.
o For instance, shoot and root fresh weight, measured at 5 days after planting are shown in Figure 1. The same parameters measured at 13 days after planting are shown in Figure 5, along with leaf area and root-to-shoot weight ratio, which are not at all provided for 5 days after planting. This looks like a total mess. It is unclear why the data for the same parameters, measured at 5 and at 13 days after planting are not provided together in the same Figure, and also, why leaf area and root-to-shoot ratio are given only for 13 days after planting.
o The data provided in Figure 2 should be provided at the beginning of the Results section (thus, the current Figure 2 should be renumbered as Figure 1) because water content in the soil sets the stage for explaining the presence or absence of changes observed in the plants at 5 and at 13 days after planting.
o The narration of the results within the main text of the Results section should follow the same order in which the data are given in the Figures. Currently, the data from all the Figures are narrated only for the non-inoculated treatments first, and then the Authors jump back to Figure 1 to narrate the results about the microbe-inoculated plants. This makes the text very difficult to follow for the reader.
o The Figure captions are written in a very sloppy manner, and they look as if the Authors haven't ever read the text after writing it. They are full of typos, nonsensical sentences, and similar. Also, as I already said, lowercase p and lowercase n should be always written in italic letters. Please thoroughly revise all the Figure captions.
o line 178-179: Please rephrase to: "the average fresh weight of shoots and roots was 19% and 25% lower, respectively, in plants planted in groups of three, compared to individually planted plants".
o line 203: Are you talking about the response to actual shading here, or about a preventive mechanism that is supposed to act as a response to the impending threat of shading? Please be clear and specific.
o line 206: Which factor? Did you mean overly extensive energy investment in chlorophyll synthesis? Please be specific in your statements.
o line 210: What is "bacterial treatment" here? We do know what it means, but a figure caption is meant to stand alone and to be clear just from reading it without having to read the article, or even the Abstract text.
o line 223-231: This is discussion text, it for sure does not belong to the Results section.
o line 274: Please correct "(B)" to "(C)"
· Discussion - The Discussion section needs thorough revision of scientific language, as well as more robust discussion of two important points that remain unaddressed:
o The Authors suggest that the use of ABA-degrading bacteria is beneficial to plant biomass yields to reduce growth inhibition from dense plantations. This is indeed a very valuable conclusion, and if just for this, your work can be considered very valuable. However, as I pointed out before, it is likely that these findings apply only to environmental conditions in which plants are not exposed to abiotic stress, at least to intense abiotic stress. Since you have not investigated how the use of P. plecoglossicida affects plant viability under abiotic stress, you need to strongly emphasize this limitation of your research both within the Discussion and Conclusions sections of your manuscript, as well as within the Abstract. You might also suggest that future research should investigate into that direction, at least it would be interesting to estimate at which intensity of abiotic stress the use of ABA-degrading bacteria can become counter-productive to plant yields.
o In your paper, you have not provided a satisfactory mechanistic explanation for the degradation of ABA by P. plecoglossicida, especially with regards to the location where this degradation occurs. In this sense, I find the results displayed in Figure 7 extremely interesting, but totally unexploited in your Discussion. The ABA content is elevated in the soil pots and in shoots, but not in roots of the non-inoculated densely planted plants compared to the soil pots and shoots of all other experimental treatments. Apparently, this elevation of ABA both in the soil pots, and in the shoots, seems to be well-mitigated through the inoculation of the densely planted plants with Pseudomonas plecoglossicida. What do these data tell us? Where does the degradation of ABA actually occur? It occurs in the soil pots, obviously, but how do the bacteria affect the ABA levels in plant shoots? Do the bacteria migrate through the root system and enter plant shoots to endogenously degrade the ABA within plant shoots? (Hopefully not.) Why is ABA present in the rhizosphere in the first place, and how does its degradation in the rhizosphere affect its endogenous levels in shoots? Is it possible that densely planted plants exude such substantial amounts of ABA into the rhizosphere when they are densely planted? If so, do they take it up with their root system and transport it to their aboveground parts in amounts extensive enough to significantly affect their endogenous shoot content? All these questions remain unanswered in your Discussion section, for which the mechanistic explanation of your findings is missing. You should thoroughly amend your Discussion section with this kind of discussions, and please take care to extensively support it with relevant findings published in the literature.
o line 291: How do you mean "according to preliminary data"? I think I might have seen, within this very paper, your results regarding the growth of P. plecoglossicida on nutrient media with ABA as the only carbon source. Why do you call these data preliminary? Please either explain or revise.
I believe that the weaknesses of your paper can be successfully addressed and overcome, and if that happens, your paper might become a valuable contribution to the readership of Biomolecules. However, for that to happen, the Authors must thoroughly and seriously address my remarks.
Kind regards,
Reviewer
Comments on the Quality of English LanguageThe English language of the manuscript requires moderate revision. However, the use of scientific language is very bad. I urge the Authors to perform an extensive revision of the scientific language of the entire manuscript by a senior scientist, preferably one who has extensive working experience from a lab abroad.
Author Response
Reviewer’s comments: After reading your manuscript, I can tell that your results are interesting and they raise interesting and useful implications. However, the manuscript suffers from multiple weaknesses, among which I would point out a very sloppy presentation of results, a weak discussion, and scientifically inappropriate language as the most important ones. For more detailed remarks, please consult my comments as given below:
General response: We are most grateful to the reviewer for such a careful analysis of our article and numerous valuable comments. We made our best to follow them. We tried to improve our “scientific language”. We modified many sentences using examples provided by respected reviewer and those from our joint article with Sally Wilkinson (known researcher, whose first language is English) published in Plant Cell Environment and cited in the present article. We also tried to find appropriate phrases in decent articles of Pubmed collection. As a result, Result and Discussion were greatly modified. Usually I provide changed sentences in responses to reviewers to make it easier for them to follow corrections of the text. However, so many changes of the original text have been made that nothing is left, but to follow them throughout the text. They can be tracked.
- Manuscript title- Please spell out 'abscisic acid' fully in the manuscript title ("Abscisic acid-degrading bacteria...")
Response: It is abscisic acid now
- Scientific and English language- The English language of the manuscript requires moderate revision. However, the use of scientific language is very bad. I urge the Authors to perform an extensive revision of the scientific language of the article. I will give you some examples below, but I urge you to ask for revision of the scientific language of the entire manuscript by a senior scientist, preferably one who has extensive working experience from a lab abroad.
Response: The revised article was mostly corrected by our colleague, whose first language is English (former Chief Editor of Annals of Botany)
3.The bacteria do not destroy abscisic acid, they degrade it. Please thoroughly revise throughout the manuscript text, using the "Find/replace" tool (lines 18, 26, 64, 156, 267, 289, 305, 309; possibly elsewhere as well)
Response: We are sorry for using inappropriate term. According to the remark of respected reviewer, it was substituted with “degrade”, “degradation” throughout the whole text
- lines 38 and 41: Please replace "was" with "is"
Response: “Was” was replaced with “is”
- line 58: carotenoids
Response: spelling of this word was corrected
- lines 66-67 and 292: ABA served as the only carbon source
Response: ABA as the only carbon source is now mentioned on lines 66-67 and 292 in the revised version of the article.
- line 82: either individuallyor in groups of three
Response: It is now “individually or in groups of three”
- line 101: indole-3-acetic acid
Response: in the revised variant it is indole-3-acetic acid. Thanks for correction
- the lowercase letters p(for confidence interval) and n (for sample size) should always be written in italic letters. Please correct in line 174 and within all figure captions.
Response: lowercase letters p (for confidence interval) and n (for sample size) are now written in italic letters in the revised version. Thanks. Sorry for having not paid attention to this previously.
- line 193: What is "the action of increased plant density"? Kindly revise to meet the standards of scientifically appropriate language
Response: this was substituted with “at the beginning of competition experiment”. We hope this might be better. We have taken the new variant from our article with Sally Wilkinson published in Plant Cell Envinronment in 2011 (cited in the present article)
- line 205: Between the treatments, not "between the options"
Response: “options” were substituted with “treatments”
- Figure 4 (both the figure caption and the figure itself): square centimeter should be written as cm2, not "cm2" nor "sm2".
Response: We are sorry for this silly mistake. It was corrected. Thanks for drawing our attention to this.
- line 236-237: "accumulation of shoot weight" - please revise to meet the standards of scientifically appropriate language
Response: this passage now goes as “accumulation of shoot biomass”
- line 250: Please always use English language
Response: Something was obviously missing from the phrase in this line. So we decided to use the phrase from our original article published in Physiol. Plant: “the substrate solution, consisting of o-phenylene-diamine (1 g L-1): 0.3 M phosphate buffer pH 5.8: 3% hydrogen peroxide in the ratio of 10 ml: 15 ml: 50 µl, was added”. We hope it is better now.
- line 263: There is no such thing as a "physic-chemical assay". Please check online the correct way to translate a word into English before you use it in your paper.
Response: it is now “physicochemical assay”
- line 282: The nitrogen index cannot be "the same", since the treatment is not the same. It is not even equal, let alone same. I believe that you wanted to say that they are not statistically significantly different from each other.
Response: Thanks! It is now as you advised: “nitrogen index in single and grouped plants was not statistically different from each other”
- line 294: Please do not call ABA "this substance". Replace it with a more appropriate term, such as "this growth regulator".
Response: “substance” is substituted with “growth regulator”
- line 325: What is "transpient"? Did you mean "transient"?
Response: this mistake is corrected.
- Introduction- Apart from language imperfections, the Introduction section is virtually flawless. The problem background is correctly narrated and naturally leads the reader towards the very well-defined research goal. However, it is difficult to read because it is presented as a single block of text. Division of the introduction into separate paragraphs would be necessary to suggest to the reader what are the logical parts of the Introduction. For instance, a new paragraph is needed starting from line 55 ("Although the approach..."), when the reader gets to know why natural alternatives to ABA inhibitors are needed. Similarly, paragraph separators should be introduced elsewhere in the Introduction to suggest the logical structure of the Introduction to the reader. I recommend the division of the Introduction section into 3-4 paragraphs in total.
Response: Thanks for this advice. In accordance, we divided introduction into several paragraphs. The first of them suggests that, along with resource scarcity, there is probably some other factor that inhibits the growth of competing plants. The second paragraph states that ABA, acting as a signal of the presence of neighbors, can contribute to inhibiting plant growth, and reducing its concentration in competing plants can have a beneficial effect. The third paragraph presents information on the origin of ABA in soil and its uptake by plants. And in the forth paragraph, the prospects for using an inhibitor of ABA synthesis and microbes that degrade ABA are compared in favor of the latter. We also added a paragraph on general interaction of plants and plant growth promoting bacteria. The last paragraph just describes the aim of research
- Experimental design - Your experimental design is well-planned and correct. I would say that it does fulfill minimal requirements for a scientific publication. I would be happier if your experimental design was more thorough and comprehensive (please see below) because a more thorough experimental design would make the evidence for your claims more compelling and more scientifically sound. However, I still find your findings and conclusions scienifically valid and sufficient for a (modest) publication. Nevertheless, I want to share with you my thoughts about what additions I would have liked to see in your experiment, so that you might plan your experimental designs more thoroughly in your future research.
Response: Thanks for your advices. We shall try to follow them if not in the present article, then in the future.
- Planting densities- I would have liked to see a broader variety of "planting densities" instead of just 1 and 3 plants per pot. If you had used 1, 2, and 3 plants per pot, or 1, 3, and 5 plants per pot in bigger pots, it would be interesting to see whether you would have observed obvious trends, or if there would be a degree of randomness to the measured data. Your results obtained with 1 and 3 plants per pot are still valid anyway, and can be considered sufficient for drawing conclusions - but more treatments always gives more information.
Response: This planting density was chosen based on our previous research published in Plant Cell Environment (2011), Plant Growth Regul (2017) and J.Plant Physiol. (2018) (cited in the present article). We are afraid that five-fold increase in planting density might be too much for bacteria to improve plants growth. But we shall try next time. Thanks for advice.
- Abiotic stress response - A more serious remark regarding your experimental design concerns the ABA-mediated response of plants to abiotic stress. Everyone can agree that the ABA-mediated growth inhibition in response to dense planting is counter-productive in agronomical terms in the absence of abiotic stress. However, the biological sense of this response results from plant evolution in conditions where harsh abiotic stress frequently occurred. Under abiotic stress, the ABA regulatory module is switched on, promoting a slow-down in plant metabolism and growth, in favor of energy saving and allocation of resources to the activation of tolerance mechanisms. For that reason, in stress conditions, the ABA regulatory module also downregulates other phytohormonal modules, that are related to growth and biomass accumulation in optimal environmental conditions (such as the one regulated by cytokinin, for instance, see Liu et al. 2013: https://doi.org/10.1186/1471-2164-14-594). In this light, the use of ABA-degrading microbes is likely to become counter-productive when the plants are subjected to abiotic stress, because they might negatively interfere with the adaptive plant responses to abiotic stress. Having this in mind, I would have liked to see, among your treatments, how plant growth (or survival) would be affected by the addition of Pseudomonas plecoglossicida when the plants are subjected to abiotic stress (for instance, drought, freezing, salinity, heat stress...). You might want to address this question in your next research.
Response: thanks to the respected reviewer for raising such an important question. In accordance we added following paragraph to the Introduction (“The accumulation of ABA in soil can, in turn, influence plant resistance to water shortage, since ABA accumulation during water stress maintains plant water status by promoting stomatal closure [Dodd 18]. ABA can however also inhibit root and shoot growth of well-watered plants [Sun 19; Li, 20]”) and to the Discussion: Although bacteria-induced decline in ABA concentrations in competing plants produced beneficial effects in the present experiments, the results of bacterial inoculation may be different under stress conditions. Our recent experiments have shown that treatment of barley plants with another bacterial strain increased their salt resistance due to bacteria-induced accumulation of ABA in the roots which strengthened apoplastic barriers thereby preventing penetration of toxic ions [37]. It will be important to check the effects of ABA degrading bacteria on plant salt resistance.
Materials and Methods - The M&M section is mostly well-written, but occasionally certain clarifications are needed:
- line 108: Please add clarification: "was added to the growing pots one day after the seedlings were transplanted, to yield..."
Response: The passage was modified in attempt to make it clearer: One day after planting, one and three plants per pot, a suspension of cells of Pseudomonas plecoglossicida 2.4-D strain… was introduced
- line 108: There is a major issue with your article regarding the microbial density in the two bacterial treatments.In line 108, you claim that the two treatments consisted of 3.8 x 106 and 3.8 x 108 CFU per g of soil. However, in the rest of the manuscript, you refer to these treatments as "106" and "108", and the 3.8 factor is lost. This can be addressed if you corrected the names of the treatments to "3.8 x 106" and "3.8 x 108" everywhere throughout the manuscript, including all Figures, and Figure captions. This is a lot of work, and also might make the text more difficult to read. A more simple solution might be to add a clarification to the text in lines 108-109: "8 x 106 and 3.8 x 108 CFU per g of soil, hereafter designated as the '106' and '108' treatments, respectively".
Response: Thanks for your nice last suggestion. We followed it and introduced “thereafter designated as the '106' and '108' treatments, respectively”
- line 111: Please add a clarification: "the freshweight of shoots and roots"
Response: We added that fresh weight of shoots and roots were measured immediately after harvesting
- line 114: Please provide a brief explanation about the meaning of the NBI index, so that the readers don't need to look for the source reference to read about it. This is very important, since in your paper, you draw important conclusions from your measurements of NBI, so it is necessary that the readers are able to readily grasp its meaning.
Response: In accordance we added that “Nitrogen balance index (NBI), calculated as the ratio between chlorophyll and flavonols contents has been shown to be a very sensitive indicator of crop N status”.
- line 118: Please fully spell out "Abscisic Acid" in the title of the subsection 2.4.Abbreviations should be fully spelled out in section and subsection titles even though they have been spelled out in the manuscript text previously.
Response: In accordance with this remark we changed the title of the article. It is now as follows: “The growth-inhibitory effect of increased planting density can be reduced by abscisic acid degrading bacteria.” ABA is also substituted with abscisic acid in the section titles.
- line 144: Which antisera? Please specify.
Response: in the revised version it is now “antisera against ABA”
- line 166 (and possibly elsewhere as well): Please always write "L", "mL", and "µL" with a capital L.
Response: It is L everywhere now.
- Section 2.6: Which software was used for statistical analysis?Please fully spell out the software name, company name, city, and country.
Response: We added that Statistica version 10 (Statsoft, Moscow, Russia) was used
- Results- The Results section of your manuscript is in a very sloppy state, for which it is in sharp contrast with some other parts of your manuscript, such as Introduction, which is very tidy. The scientific language is in its poorest shape in the Results section compared to the rest of the manuscript, and overall, it looks like this section has not been revised by a senior scientist. Also, the results are displayed in chaotic order both in the Figures, and in the narration of the results within the manuscript text.
Response: We have done our best trying to improve our “scientific language”. We modified many sentences using examples provided by respected reviewer and those from our joint article with Sally Wilkinson (known researcher, whose first language is English) published in Plant Cell Environment and cited in the present article. We also tried to find appropriate phrases in decent articles of Pubmed collection. As a result, Result and Discussion were greatly modified. Usually I provide changed sentences in responses to reviewers to make it easier for them to follow corrections of the text. However, so many changes of the original text have been made that nothing is left, but to follow them throughout the text. They can be tracked.
- For instance, shoot and root fresh weight, measured at 5 days after planting are shown in Figure 1.The same parameters measured at 13 days after planting are shown in Figure 5, along with leaf area and root-to-shoot weight ratio, which are not at all provided for 5 days after planting. This looks like a total mess. It is unclear why the data for the same parameters, measured at 5 and at 13 days after planting are not provided together in the same Figure, and also, why leaf area and root-to-shoot ratio are given only for 13 days after planting.
Response: According to the recommendation of respected reviewer, we joined the data showing the results obtained on the 5th and the 13th day, which are now presented in Figure 3. We also provided the data showing root-to-shoot weight ratio and leaf area obtained on the 5th day in figure 4.
- The data provided in Figure 2 should be provided at the beginning of the Results section (thus, the current Figure 2 should be renumbered as Figure 1)because water content in the soil sets the stage for explaining the presence or absence of changes observed in the plants at 5 and at 13 days after planting.
Response: Thanks for valuable comment. Now figure showing dynamics of water content goes first. We also put the data showing nitrogen index just after it so that to show when plants began to suffer from water and nitrogen shortage
- The narration of the results within the main text of the Results section should follow the same order in which the data are given in the Figures.Currently, the data from all the Figures are narrated only for the non-inoculated treatments first, and then the Authors jump back to Figure 1 to narrate the results about the microbe-inoculated plants. This makes the text very difficult to follow for the reader.
Response: Succession of description of the data was changed and now effects of bacteria are described immediately after the effects of competition in non-inoculated plants.
- The Figure captions are written in a very sloppy manner,and they look as if the Authors haven't ever read the text after writing it. They are full of typos, nonsensical sentences, and similar. Also, as I already said, lowercase p and lowercase n should be always written in italic letters. Please thoroughly revise all the Figure captions.
Response: We worked hard and have done our best trying to improve the Figure captions. They are now as follows:
Figure 1. Dynamics of water content of sandy soil in pots containing individual plants or groups (three per pot), expressed as a % of total water capacity, measured immediately before watering. Rhizosphere was inoculated with bacteri a (10⁶ and 10⁸ CFU/g) one day after planting. n=12.
Figure 2. Nitrogen index (NBI) measured after 5 (A) and 13 (B) days of growing lettuce plants individually or in groups of three. Rhizosphere was inoculated with bacteria (10⁶ and 10⁸ CFU/g) one day after planting. Means ± SE are presented. Statistically different means (n = 40) are indicated by different letters (p < 0.05, ANOVA, Duncan’s test).
Figure 3. Shoot (A, C) and root (B, D) fresh weight measured after 5 (A,B) and 13 (C,D) days of growing lettuce plants individually or in groups of three. Rhizosphere was inoculated with bacteria (10⁶ and 10⁸ CFU/g) one day after planting. Means ± SE are presented. Statistically different means (n = 12) are indicated by different letters (p < 0.05, ANOVA, Duncan’s test).
Figure 4. Leaf area (A,C) and r fresh weight ratio (B, D) measured after 5 (A,B) and 13 (C,D) days of growing lettuce plants individually or in groups of three. Rhizosphere was inoculated with bacteria (10⁶ and 10⁸ CFU/g) one day after planting. Statistically different means (n = 12) are indicated by different letters (p < 0.05, ANOVA, Duncan’s test).
Figure 5. Chlorophyll content (µg/cm2) measured after 5 (A) and 13 (B) days of growing lettuce plants individually or in groups of three. Rhizosphere was inoculated with bacteria (10⁶ and 10⁸ CFU/g) one day after planting. Statistically different means (n = 40) are indicated by different letters (p < 0.05, ANOVA, Duncan’s test).
Figure 6. Chromatographic profiles of ABA ((5.58±0.03) min release time) in the control (without bacteria) (A), and in the culture media of P. plecoglossicida 2.4-D (B). C - dynamics of ABA content in the control medium and in the medium with bacteria (C), n = 9.
Figure 7. ABA content per g fresh weight of shoots (A) and roots (B) and per pot with sand soil (C), measured after 13 days of growing lettuce plants individually or in groups of three. Rhizosphere was inoculated with bacteria (10⁶ and 10⁸ CFU/g) one day after planting. Statistically different means (n = 12) are indicated by different letters (p < 0.05, ANOVA, Duncan’s test).
- line 178-179: Please rephrase to: "the average fresh weight of shoots and roots was 19% and 25% lower, respectively, in plants planted in groups of three, compared to individually planted plants".
Response: The sentence was modified in the revised article: “the fresh weight of shoots and roots of each of competing plants were lower than those of plants grown alone”
- line 203: Are you talking about the response to actual shading here, or about a preventive mechanism that is supposed to act as a response to the impending threat of shading? Please be clear and specific.
Response: Thanks for this remark. We changed the sentence and it now sounds as follow: “Grouped plants showed a tendency to increase chlorophyll content compared to single plants (Figure 4A), which was probably a compensatory reaction to a signal of impending shading as planting density increased”
- line 206: Which factor?Did you mean overly extensive energy investment in chlorophyll synthesis? Please be specific in your statements.
Response: The sentence was modified: “Disappearance of the difference in the pigment content excludes changes in chlorophyll from among the possible causes of plant growth inhibition at increased planting density”
- line 210: What is "bacterial treatment" here?We do know what it means, but a figure caption is meant to stand alone and to be clear just from reading it without having to read the article, or even the Abstract text.
Response: Capture of Figure 5 was modified: “Chlorophyll content (µg/cm2) measured after 5 (A) and 13 (B) days of growing lettuce plants individually or in groups of three. Rhizosphere was inoculated with bacteria (10⁶ and 10⁸ CFU/g) one day after planting. “
- line 223-231: This is discussion text, it for sure does not belong to the Results section.
Response: sentences that belong to discussion were transferred to this section and in Results we left only that “Four days after the start of experiments, we detected no effect of either competition or inoculation on the root-to-shoot fresh mass ratio (Figure 4B). However later on (on the 13th day) the ratio of the weight of roots and shoots increased by one and a half times (from 0.45 to 0.62 in single and grouped plants, respectively (Figure 4D)
- line 274: Please correct "(B)" to "(C)"
Response: This was corrected
- Discussion - The Discussion section needs thorough revision of scientific language, as well as more robust discussion of two important points that remain unaddressed:
The Authors suggest that the use of ABA-degrading bacteria is beneficial to plant biomass yields to reduce growth inhibition from dense plantations. This is indeed a very valuable conclusion, and if just for this, your work can be considered very valuable. However, as I pointed out before, it is likely that these findings apply only to environmental conditions in which plants are not exposed to abiotic stress, at least to intense abiotic stress. Since you have not investigated how the use of P. plecoglossicida affects plant viability under abiotic stress, you need to strongly emphasize this limitation of your research both within the Discussion and Conclusions sections of your manuscript, as well as within the Abstract. You might also suggest that future research should investigate into that direction, at least it would be interesting to estimate at which intensity of abiotic stress the use of ABA-degrading bacteria can become counter-productive to plant yields.
Response: As we have written in our response to the comment above on this theme, this is a very interesting problem. We tried to solve it by adding to the Discussion the following paragraph already mentioned above: Although bacteria-induced decline in ABA concentrations in competing plants resulted in beneficial effects in the present experiments, the results of bacterial inoculation may be different under stress conditions. Our recent experiments have shown that treatment of barley plants with another bacterial strain increased their salt resistance due to bacteria-induced accumulation of ABA in the roots bringing about increased formation of apoplast barriers thereby preventing penetration of toxic ions [51]. It will be important to check the effects of ABA degrading bacteria on plant salt resistance.
- In your paper, you have not provided a satisfactory mechanistic explanation for the degradation of ABA by plecoglossicida, especially with regards to the location where this degradation occurs. In this sense, I find the results displayed in Figure 7 extremely interesting, but totally unexploited in your Discussion.The ABA content is elevated in the soil pots and in shoots, but not in roots of the non-inoculated densely planted plants compared to the soil pots and shoots of all other experimental treatments. Apparently, this elevation of ABA both in the soil pots, and in the shoots, seems to be well-mitigated through the inoculation of the densely planted plants with Pseudomonas plecoglossicida. What do these data tell us? Where does the degradation of ABA actually occur? It occurs in the soil pots, obviously, but how do the bacteria affect the ABA levels in plant shoots? Do the bacteria migrate through the root system and enter plant shoots to endogenously degrade the ABA within plant shoots? (Hopefully not.) Why is ABA present in the rhizosphere in the first place, and how does its degradation in the rhizosphere affect its endogenous levels in shoots? Is it possible that densely planted plants exude such substantial amounts of ABA into the rhizosphere when they are densely planted? If so, do they take it up with their root system and transport it to their aboveground parts in amounts extensive enough to significantly affect their endogenous shoot content? All these questions remain unanswered in your Discussion section, for which the mechanistic explanation of your findings is missing. You should thoroughly amend your Discussion section with this kind of discussions, and please take care to extensively support it with relevant findings published in the literature.
Response: Thanks for this important recommendation. In accordance with this remark we added a paragraph to Discussion:
Plants secrete organic compounds, including hormones, through their roots [47, 48]. The outflow of ABA is supported by transporters located in the cells of root epidermis [16] which gradually increase its concentration in the soil during the growing season [17]. Therefore, the increased concentration of ABA in the pots with grouped plants is not surprising, given that not one, but three plants released ABA into the rhizosphere. In turn, many substances are taken up by roots from the rhizosphere, including ABA present in the soil solution, resulting in an equilibrium between ABA concentrations in soil and plants [17]. This circulation of ABA out of and into plants may explain the higher concentration of ABA both in the sand solution and in the shoots of grouped plants. In turn the presence of bacteria capable to degrade ABA in the rhizosphere obviously reduces its concentration not only in the sand solution, but also in plants. Similar pattern was found in the case of bacteria capable of degrading ACC (a precursor of ethylene). Presence of these bacteria in rhizosphere decreased ethylene production in the plants [49]. Similar explanation may be applied to our experiments where the presence of ABA degrading bacteria in the rhizosphere was accompanied by a decline in ABA concentration in the shoots of the competing plants. Ability of bacteria to influence ABA concentration in plants might be also due to their effects on metabolism of this hormone in plants themselves. Effects of another bacterial strain on expression of the genes involved in the control of ABA metabolism in plants have been shown by us recently [35]. It remains a mystery how ABA accumulates in the shoots of grouped plants, while its concentration in the roots does not change. As mentioned above, the increase in planting density results in an increase in the pH of soil and xylem sap contributing to ABA dissociation and increased effectiveness of its transport from soil to roots and from roots to shoots [21]. Dissociation of ABA decreases its uptake by cells and thereby ABA is more readily transported through apoplast on its way from roots to shoots [Sauter 21]. This may well explain the greater accumulation of ABA in the shoots of competing plants. Still, the distribution of ABA in plants and mechanisms regulating its transport remain an important but not fully understood process. It is hoped that further study of the newly discovered ABA transporters will help to unravel this problem [50].
- line 291: How do you mean "according to preliminary data"?I think I might have seen, within this very paper, your results regarding the growth of plecoglossicida on nutrient media with ABA as the only carbon source. Why do you call these data preliminary? Please either explain or revise.
Response: We just deleted “according to our preliminary data”. This ability of the strain to grow on nutrient media with ABA as the only carbon source has been discovered prior to all other results. But since these results have never been published this “preliminary” has no sense.
General remark of the reviewer: I believe that the weaknesses of your paper can be successfully addressed and overcome, and if that happens, your paper might become a valuable contribution to the readership of Biomolecules. However, for that to happen, the Authors must thoroughly and seriously address my remarks.
Response: thanks for your kind words.
Reviewer 2 Report
Comments and Suggestions for Authors
The manuscript of Vysotskaya at al. deals with an original approach to optimize plant growth in conditions of high planting density. The authors propose the use of a bacterial strain that could modulate the levels of ABA in plant tissues and root exudates and mitigate the ABA-mediated plant growth repression. The Pseudomonas strain has been shown to effectively reduce the content of ABA in both the cell suspension medium and the plant growth substrate. Interestingly, the treatment with bacterial cell suspension has also resulted in repressed accumulation of endogenous ABA in shoots of densely cultivated lettuce plants. By running analyses in a time course, the authors have demonstrated that, at least at the early stages, the negative impact of plant competition on growth-related traits is related to increased ABA levels rather than water and nutrient limitation. As a whole, this report is a valuable contribution to the efforts for delivering novel strategies for increasing crop yield per arable area.
Remarks:
1. Vysotskaya at al. have reported a statistically significant retention in shoot ABA levels after treatment with the Pseudomonas strain when lettuce plants have been grown at high density. I believe this finding needs further attention, at least in the Discussion section. The reduction in the ABA concentration in the sand upon addition of bacterial cells can be regarded as a direct effect related to their metabolic activity. However, the ABA decrease in plant tissues that are not in direct contact with the bacterial cells remains puzzling. It is likely that the Pseudomonas-derived IAA released to the growth substrate might influence the host plant metabolism.
2. The dynamics in endogenous ABA content is intrinsically related to plant stress tolerance, especially in conditions of drought stress. As the bacterial cell suspension interferes with ABA content in case of dense planting, the authors should also assess the possible negative impact of the bacterial strain on lettuce adaptation to water shortage.
3. The authors should provide additional information about the setup of the initial screening that has led to the identification of this ABA-degrading Pseudomonas strain.
4. Please provide representative images of treated and control lettuce plants when grown individually or in a dense format, at least as a supplementary information.
5. The Y-axes of the graphs in Fig. 4 and 5C are mislabeled: sm2 should be replaced by cm2.
6. Line 240: the reference to Fig. 4A is wrong.
7. Line 319: “Most of the time, every day in the present experiments” is meaningless and should be rephrased.
Comments on the Quality of English LanguageMinor corrections needed (see my comments above).
Author Response
General comment of the reviewer: The manuscript of Vysotskaya at al. deals with an original approach to optimize plant growth in conditions of high planting density. The authors propose the use of a bacterial strain that could modulate the levels of ABA in plant tissues and root exudates and mitigate the ABA-mediated plant growth repression. The Pseudomonas strain has been shown to effectively reduce the content of ABA in both the cell suspension medium and the plant growth substrate. Interestingly, the treatment with bacterial cell suspension has also resulted in repressed accumulation of endogenous ABA in shoots of densely cultivated lettuce plants. By running analyses in a time course, the authors have demonstrated that, at least at the early stages, the negative impact of plant competition on growth-related traits is related to increased ABA levels rather than water and nutrient limitation. As a whole, this report is a valuable contribution to the efforts for delivering novel strategies for increasing crop yield per arable area.
Response: We are most grateful for valuable comments, which we tried to follow
Remarks:
- Vysotskaya at al. have reported a statistically significant retention in shoot ABA levels after treatment with the Pseudomonas strain when lettuce plants have been grown at high density. I believe this finding needs further attention, at least in the Discussion section. The reduction in the ABA concentration in the sand upon addition of bacterial cells can be regarded as a direct effect related to their metabolic activity. However, the ABA decrease in plant tissues that are not in direct contact with the bacterial cells remains puzzling. It is likely that the Pseudomonas-derived IAA released to the growth substrate might influence the host plant metabolism.
Response: Thanks for this advice. In accordance we added that “Ability of bacteria to influence ABA concentration in plants might be also due to their effects on metabolism of the hormone in plants themselves. Effects of another bacterial strain on expression of the genes involeved in the control of ABA metabolism in plants have been shown by us recently [35]”.
Still there may be direct effects of bacteria on ABA concentration in plants. To discuss this possibility we added that “Plants secrete organic compounds, including hormones, through their roots [47, 48]. Therefore, the increased concentration of ABA in the pots with grouped plants is not surprising, given that not one, but three plants released ABA into the rhizosphere. In turn, many substances are taken up by roots from the rhizosphere, including ABA present in the soil solution, resulting in an equilibrium between ABA concentrations in soil and plants [17]. This circulation of ABA out of and into plants may explain the higher concentration of ABA both in the sand solution and in the shoots of grouped plants. In turn the presence of bacteria capable to degrade ABA in the rhizosphere obviously reduces its concentration not only in the sand solution, but also in plants. Similar pattern was found in the case of bacteria capable of degrading ACC (a precursor of ethylene). Presence of these bacteria in rhizosphere decreased ethylene production in the plants [49]. Similar explanation may be applied to our experiments where the presence of ABA degrading bacteria in the rhizosphere was accompanied by a decline in ABA concentration in the shoots of the competing plants.”
- The dynamics in endogenous ABA content is intrinsically related to plant stress tolerance, especially in conditions of drought stress. As the bacterial cell suspension interferes with ABA content in case of dense planting, the authors should also assess the possible negative impact of the bacterial strain on lettuce adaptation to water shortage.
Response: Thanks for this recommendation. In accordance we added to Introduction that “that “The accumulation of ABA in soil can, in turn, influence plant resistance to water shortage, since ABA accumulation during water stress maintains plant water status by promoting stomatal closure [Dodd 18]. ABA can however also inhibit root and shoot growth of well-watered plants [Sun 19; Li, 20].” We also added to the Discussion that “Although bacteria-induced decline in ABA concentrations in competing plants resulted in beneficial effects in the present experiments, the results of bacterial inoculation may be different under stress conditions. Our recent experiments have shown that treatment of barley plants with another bacterial strain increased their salt resistance due to bacteria-induced accumulation of ABA in the roots bringing about increased formation of apoplast barriers thereby preventing penetration of toxic ions [37]. It will be important to check the effects of ABA degrading bacteria on plant salt resistance.”
- The authors should provide additional information about the setup of the initial screening that has led to the identification of this ABA-degrading Pseudomonas strain.
Response: We deleted the phrase about preliminary experiments, since their results are presented in the present work for the first time. Description of other strains capable to degrade ABA will be the theme of our next article.
- Please provide representative images of treated and control lettuce plants when grown individually or in a dense format, at least as a supplementary information.
Response: Representative images are provided as a supplementary information (Figure S1).
- The Y-axes of the graphs in Fig. 4 and 5C are mislabeled: sm2 should be replaced by cm2.
Response: This was corrected
- Line 240: the reference to Fig. 4A is wrong.
Response: The numbers of figures were corrected
- Line 319: “Most of the time, every day in the present experiments” is meaningless and should be rephrased.
Response: Sorry for the awkward phrase. The sentence was modified as follows: “Although competing plants extracted more water from the pots, the increased water loss was offset by daily watering, which maintained soil moisture levels sufficient to support the growth of the inoculated grouped plants”
Reviewer 3 Report
Comments and Suggestions for Authors
This research sheds light on the potential for sustainable and eco-friendly agricultural practices by emphasizing the role of ABA-degrading bacteria in mitigating the adverse impacts of competing lettuce plants. Beyond the specific findings in the context of increased planting density, these results hold implications for the broader field of plant biology and agriculture.
Researching the complicated relationship between plant hormones and microbial interactions can help us understand how competition-induced growth inhibition works. This knowledge can inform the development of innovative strategies for optimizing crop yields and reducing the need for synthetic chemical interventions.
1.The title should change to be more specific and clear
2. 1. The level of English throughout the manuscript needs to meet the journal's standard. Therefore, you may wish to ask a native speaker to check your manuscript for grammar, style, and syntax.
2. The abstract must have a rationale, an objective, materials and methods, results, and conclusions. The first sentence must be a rationale and research Gap for this study. Therefore, the abstract should address why the study was conducted. There must be a mention of any existing research gap or complex interplay between plant hormones and microbial interactions.
3. The introduction should mention the complex interplay between plant hormones and microbial interactions. In addition, what are the mechanisms underlying competition-induced growth inhibition
4. The materials and methods could have been clearer to me, and I need help understanding what the authors did due to the incorrectly arranged data in this section.
Subtitles should be rearranged to understand the materials and methods as follows:
1. Location and environmental conditions
2. Field description and experimental design
3. Plant materials
4. Treatments
Please explain what the experimental design is, and treatments as well as the statically analysis in a clear way.
What about the period for conducting this experiment in days?
Please include the following details:
The experiment started on day month year and ended on day month year. For example, The experiment started on 22 June 2016 and ended on 22 October 2020.
Please mention why the seedling is grown in soil, not in other growth media such as perlite, peatmoss etc.
Please mention what is control treatment
Please mention the size of pots in liter
Please mention the interval irrigation
5.The discussion section must be presented under certain subtitles, as the authors did for the results. As the authors presented their results under certain subtitles in the Results section, they also suggested developing subtitles under the Discussion section. The authors mostly only make comparisons of their results with the literature's results. However, they need to discuss the mechanisms by which the results are obtained
Comments on the Quality of English Language
The level of English throughout the manuscript needs to meet the journal's standard. Therefore, you may wish to ask a native speaker to check your manuscript for grammar, style, and syntax
Author Response
Reviewer’s comments:
General comment of the reviewer:This research sheds light on the potential for sustainable and eco-friendly agricultural practices by emphasizing the role of ABA-degrading bacteria in mitigating the adverse impacts of competing lettuce plants. Beyond the specific findings in the context of increased planting density, these results hold implications for the broader field of plant biology and agriculture. Researching the complicated relationship between plant hormones and microbial interactions can help us understand how competition-induced growth inhibition works. This knowledge can inform the development of innovative strategies for optimizing crop yields and reducing the need for synthetic chemical interventions.
Response: We are most grateful to the reviewer for valuable comments. We have do ne our best to address them
1.The title should change to be more specific and clearer.
Response: The title was modified. It is now as follows: “The growth-inhibitory effect of increased planting density can be reduced by abscisic acid degrading bacteria”
- The level of English throughout the manuscript needs to meet the journal's standard. Therefore, you may wish to ask a native speaker to check your manuscript for grammar, style, and syntax.
Response: The revised text was corrected by our Colleague, whose first language is English (former Chief Editor of Annals of Botany)
- The abstract must have a rationale, an objective, materials and methods, results, and conclusions. The first sentence must be a rationale and research Gap for this study. Therefore, the abstract should address why the study was conducted. There must be a mention of any existing research gap or complex interplay between plant hormones and microbial interactions.
Response: Thanks for this valuable comment. The respected reviewer opened our eyes on the poor design of the abstract. It was completely rewritten as follows:
Abstract: High-density planting can increase crop productivity per unit area of cultivated land. However application of this technology is limited by inhibition of plant growth in the presence of neighbors, which is not only due to their competition for resources, but is also caused by growth regulators. Specifically, abscisic acid (ABA) accumulated in plants under increased density of planting has been shown to inhibit their growth. The goal of the present study was to test the hypothesis that bacteria capable of degrading ABA can reduce the growth inhibitory effect of competition among plants by reducing concentration of this hormone in plants and their environment. Lettuce plants were grown singly and three per pot and the rhizosphere was inoculated with a strain of Pseudomonas plecoglossicida 2.4-D capable of degrading ABA. Plant growth was recorded in parallel with immunoassaying ABA concentration in the pots and plants. Presence of neighbors indeed inhibited the growth of non-inoculated lettuce plants. Bacterial inoculation positively affected the growth of grouped plants, reducing the negative effects of competition. Bacteria-induced increase in the mass of competing plants was greater than that of single ones. ABA concentration was increased by the presence of neighbors both in soil and plants associated with inhibition of plant growth, but accumulation of this hormone was prevented by bacteria as well as inhibition of the growth of grouped plants. The results confirm the role of ABA in the response of plants to the presence of competitors and possibility of reducing the negative effect of competition on plant productivity with the help of bacteria capable of degrading this hormone.
- The introduction should mention the complex interplay between plant hormones and microbial interactions. In addition, what are the mechanisms underlying competition-induced growth inhibition.
Importance of bacteria for the growth of plants in general is now mentioned in the Introduction:
“The capacity of so-called plant growth promoting rhizobacteria to stimulate plant growth more generally is attracting increasing attention [26; 27; 28]. For example, microorganisms can directly influence plant growth by synthesizing growth-stimulating hormones [29; 30] and degrading growth-inhibitory hormones [31, 32]). Changes in plant hormonal status may therefore result from either microbial consumption or production of hormones, or from changes in plant hormone metabolism in planta or both (33; 34].”
This is a really exciting theme and corresponding author of this article has written a review about it published in Front. Plant Sci. (Kudoyarova et al., 2019). But we did not go too deep into this problem here, since it is slightly beyond the scope of this article.
Concerning the mechanisms underlying competition-induced growth inhibition it is said in the article that “The growth inhibition is explained not simply by competition between closely spaced plants for resources in the soil, because it occurs even before the presence of near-neighbors reduces availability of water and mineral nutrients [1, 7]. This is a consequence of the plant reaction to the environmental signals created by the close presence of neighbors [8, 9].”
- The materials and methods could have been clearer to me, and I need help understanding what the authors did due to the incorrectly arranged data in this section.
Subtitles should be rearranged to understand the materials and methods as follows:
- Location and environmental conditions and Field description
Response: We are sorry for unclear description of conditions of plant growing. We obviously failed to emphasize that experiments were conducted under controlled laboratory conditions. In the present variant of the article, this is emphasized: “Lettuce plants (Lactuca sativa L., cultivar Moscow Greenhouse) were grown under controlled laboratory conditions at an irradiance of 320 μmol m-2 s-1 PAR, a 14-hour photoperiod, 18–26 °C, and air humidity of 40–50%.”
- experimental design
Response: Experimental Design is described in corresponding section 2.1.“Lettuce seeds were germinated in the soil and after the third true leaf appeared on the seedlings, they were transplanted into 0.2 L pots (either individually or in groups of three, simulating different planting densities) with 0.2 kg of sand soaked with Hoagland-Arnon solution (5 mM KNO3, 5 mM Ca(NO3)2, 1 mM KH2PO4, 1 mM MgSO4) until completely saturated. …One day after planting, one and three plants per pot, a suspension of cells of Pseudomonas plecoglossicida 2.4-D strain (GenBank KY593189.1) from the collection of microorganisms of the Ufa Institute of Biology [35], was introduced into the rhizosphere (to the surface around each seedling).
- Plant materials
Response: We hope that enough is said about plant materials in the section about experimental design (see above)
- Please explain what the experimental design is, and treatments as well as the statically analysis in a clear way.
Response: In accordance with the remark of the reviewer we added that “In this case, two types of plant treatment were used: increased planting density (placing 3 plants in a pot instead of one) and rhizosphere inoculation with Pseudomonas plecoglossicida 2.4-D.” Regarding statistical analysis we added that “Data were processed by means of one-way analysis of variance (ANOVA) with Duncan’s multiple range tests used to discriminate means (p≤ 0.05) using Statistica version 10 (Statsoft, Moscow, Russia).”
- What about the period for conducting this experiment in days?
Response: The period of conducting experiments with plants is describe in section 2.3. : “Four and 12 days after inoculation (5 and 13 after transplantation) the fresh weight of shoots and roots (measured immediately harvesting), leaf area, pigment content, and nitrogen index of lettuce plants were determined.” Experiments also included growing bacteria for inoculation of rhizosphere and assay of ABA in culture medium supplied with this hormone. Information about duration of bacteria growth can be found in section 2.2. Bacterial Strain and Culture Media, where it is said that “To obtain the bacterial preparation, microorganisms were grown … for 2 days.” Information about this is also in section 2.5., where it is said that “To study degradation of ABA by bacteria, they were grown for 10 days in Raymond's liquid medium [42], to which ABA was added to a final concentration of 100 mg/L under conditions described in section 2.2.”
- Please include the following details:
The experiment started on day month year and ended on day month year. For example, the experiment started on 22 June 2016 and ended on 22 October 2020.
Response: As mentioned above experiments were conducted under controlled laboratory conditions. In this case the dates are not necessary
- Please mention why the seedling is grown in soil, not in other growth media such as perlite, peatmoss etc.
Response: Sand was chosen, since it is easier to sterilize it. To clarify this we added that “Prior to experiments, sand was sterilized by calcinations to exclude the presence of undesirable bacteria.”
- Please mention what is control treatment.
Response: In accordance with this remark we added that “Single non-inoculated plants served as a control.” Please mention the interval irrigation
- Please mention the size of pots in liter.
Response: The pot size is mentioned in liter in the revised variant (seedlings, they were transplanted into 0.2 L pots)
- Please mention the interval irrigation
Response: In interval irrigation is mentioned in section 2.1. The description was modified as follows to make it clearer: “Plants were watered each day to reach 70-80% of full water capacity, by adding to all pots the same volume of nutrient solution (100% H-A) as a top dressing and the required amount of water, depending on transpiration losses.”
- The discussion section must be presented under certain subtitles, as the authors did for the results. As the authors presented their results under certain subtitles in the Results section, they also suggested developing subtitles under the Discussion section.
Response: In accordance with the recommendation of respected reviewer, following subtitles have been introduced into Results and Discussion:
Results: 3.1. Effects of competition and bacterial treatment on soil humidity and plant nitrogen index
3.2. Effects of competition and bacterial treatment on the growth and chlorophyll content of lettuce plants.
3.3. Effects of P. plecoglossicida 2.4-D on concentration of abscisic acid in bacterial culture media, sandy soil and lettuce plants
Discussion: 4.1. Effects of resource availability on the growth of lettuce in the presence of neighbors and reduction of growth inhibition upon rhizosphere inoculation with P. plecoglossicida 2.4-D.
4.2. Involvement of abscisic acid in competition induced inhibition of lettuce growth.
4.2.1. Importance of the capacity of P. plecoglossicida 2.4-D to decrease concentration of abscisic acid in bacterial culture media, soil and lettuce plants.
4.2.2. Synthesis of abscisic acid by plants, its bacterial degradation and their importance for maintaining hormonal balance in the plant/rhizosphere system
- The authors mostly only make comparisons of their results with the literature's results. However, they need to discuss the mechanisms by which the results are obtained.
Response: In accordance with this comment, it was added to the Discussion that “Plants secrete organic compounds, including hormones, through their roots [47, 48]. Therefore, the increased concentration of ABA in the pots with grouped plants is not surprising, given that not one, but three plants released ABA into the rhizosphere. In turn, many substances are taken up by roots from the rhizosphere, including ABA present in the soil solution, resulting in an equilibrium between ABA concentrations in soil and plants [17]. This circulation of ABA out of and into plants may explain the higher concentration of ABA both in the sand solution and in the shoots of grouped plants. In turn the presence of bacteria capable to degrade ABA in the rhizosphere obviously reduces its concentration not only in the sand solution, but also in plants. Similar pattern was found in the case of bacteria capable of degrading ACC (a precursor of ethylene). Presence of these bacteria in rhizosphere decreased ethylene production in the plants [Glick 49]. Similar explanation may be applied to our experiments where the presence of ABA degrading bacteria in the rhizosphere was accompanied by a decline in ABA concentration in the shoots of the competing plants. Ability of bacteria to influence ABA concentration in plants might be also due to their effects on metabolism of this hormone in plants themselves. Effects of another bacterial strain on expression of the genes involved in the control of ABA metabolism in plants have been shown by us recently [35]. It remains a mystery how ABA accumulates in the shoots of grouped plants, while its concentration in the roots does not change. As mentioned above, the increase in planting density results in an increase in the pH of soil and xylem sap contributing to ABA dissociation and increased effectiveness of its transport from soil to shoots [21]. This may well explain the greater accumulation of ABA in the shoots of competing plants. Still, the distribution of ABA in plants and mechanisms regulating its transport in plants remain an important but not fully understood process. It is hoped that further study of the newly discovered ABA transporters will help to unravel this problem [50].”
Round 2
Reviewer 1 Report
Comments and Suggestions for Authors
Dear Authors,
I have performed the second round of review for your manuscript "The growth-inhibitory effect of increased planting density can be reduced by abscisic acid degrading bacteria", submitted for publication in Biomolecules.
I can see that your manuscript has been extensively revised since the first submission. All of my remarks have been seriously addressed, and the manuscript has now greatly improved in all the three areas (tidyness of Results, scientific robustness of the Discussion, the use of scientific language) that I had identified as problematic during the first round of review. The manuscript can now be considered a valuable contribution to plant biology, however a final round of minor corrections is needed before the article is ready for publication in Biomolecules. For the final round of corrections, please follow my comments as given below:
· Manuscript title: please add a dash between "acid" and "bacteria" ("abscisic acid-degrading bacteria"). Please also make the same correction elsewhere in the manuscript, as appropriate.
· line 23: please correct "in soil and plants" into "in soil and plant shoots", as Figure 7 shows, that the increase in endogenous ABA was found only in shoots, but not in roots of the densely planted plants.
· line 24: please re-organize the order of words in this sentence ("but accumulation of this hormone, as well as inhibition of the growth of grouped plants, was prevented by bacteria").
· line 57: please replace the citation of Vysotskaya et al. 2011 with the reference number
· line 58: please add a comma after "xylem sap"
· line 78: please provide the reference immediately after this sentence, even though it might be a repetition of a reference cited immediately previously or afterwards. Currently it is not obvious which reference this text refers to.
· line 83: their ABA concentration in which tissue? And please provide the reference again immediately after this sentence.
· line 198: please put p in italic here too.
· line 202: please add the article ("containing a single plant")
· line 206: please delete the word "occasion". Or you can replace it with "watering event" as you wrote afterwards. "Watering event" may sound appropriate, but "occasion" is not really a good word.
· line 206: also please add a comma after "capacity"
· line 210: I see another interesting thing in this graph, that might be worth a comment. The soil water content seemed to additionally decrease in the 108 treatment (compared to both non-inoculated plants, and 106-inoculated plants) in both individually and densely planted plants. How would you explain this? I see two possible explanations: (1) the higher bacterial inoculum might have stimulated the plants to extract the water from the soil more efficiently than they did in the absence (or presence of lower inoculum) of bacteria; or (2) it might be that the bacteria themselves extracted the water from the soil when inoculated at higher density. Although I'm not sure whether bacteria would have sufficient biomass to do that on a measurable scale. Or perhaps they do, if this strain forms a biofilm? In any case, I would like to read a commentary on this phenomenon, and I think it can be added directly to the Results section (not even to the Discussion), because it would serve more as a direct commentary on the displayed result, than as a discussion which is tightly related to your main research goal.
· Figure legends (for all Figures): Since you decided to stick to the labels "106" and "108" throughout the manuscript text (instead of "3.8 x 106" and "3.8 x 108") I would prefer that you just write "106" and "108", because mentioning the measurement unit "CFU/g" suggests the exact density, which would be incorrect.
· line 222: Here you use the British English spelling "neighbours", whereas I believe that in most of the rest of the manuscript you chose American English spelling for "neighbors", and for most other words. Please check the manuscript thoroughly for the consistent use of either British or American English.
· line 233-234: You doubled the time designation ("by this time/at this time"). Please remove one of them.
· line 237: Please, again delete the measurement unit "CFU/g" (or add the 3.8 x factor before the number).
· line 256: Please add "of inoculum density" ("the effect of inoculum density")
· line 269, 275, 316: Why four days? I believe it should be five days.
· line 269-271: It is necessary to emphasize, together with the sentence citing Figure 4A, that this effect was statistically significant in the non-inoculated plants, while in the inoculated plants it was observed too, but was not confirmed as statistically significant.
· line 274: Please add "leaf" ("between the leaf weight of single and competing plants")
· line 276-278: The statement citing Figure 4D is incomplete or inaccurate. You wanted to say that at day 13, it was recorded that the competition factor, in non-inoculated plants, increased the root-to-shoot ratio (an increase from 0.45 to 0.62 is not a 1.5-fold increase, please recalculate). You should also emphasize that this effect of competition on the root-to-shoot ratio, was decreased by inoculation, so the difference between single and competing plants was less pronounced in inoculated plants.
· line 280-281: In the sentence citing Figure 5A, it should be added that this relationship, however, was confirmed by Duncan's test as statistically significant only at the 106 treatment.
· Figure 7: Please add directly to the figure (either on the y-axis or as a title above each subfigure) the designations "shoots", "roots", "soil" so that the reader does not have to scroll down to read the figure caption to remember what each graph is about.
· Discussion – Discussion is really greatly improved and I want to congratulate you on the new version of the Discussion, which I like very much. I also welcome the fragmentation of the Discussion section into subsections, however I find that the subsection titles are too long and complicated, please shorten them down to make them easier to grasp
· line 321 – "detected in our experiments at the beginning of the competition" – this statement is inaccurate. The second part of this same statement (correctly) states that there was an absence of change in this indicator at the beginning of the experiment.
· line 328-329 – this sentence too is not completely precise, even though it is not really incorrect. The plants indeed did not suffer from nitrogen deficiency, but the soil water content was even lower in inoculated than in non-inoculated plants – which might mean that the water uptake was more efficient in inoculated plants, which may be the cause why they did not have to increase their investment in root growth compared to single plants. Please make these statements more detailed and clear.
· line 350-351: At this point, the readers might be interested to get to know briefly about how Rhodococcus (by which mechanism) was able to reduce the endogenous levels of ABA, i.e., in which plant tissues.
· line 362-364: This sentence is too complicated and very difficult to follow, please make it easier to read.
· line 371-373: It is unclear whether this sentence refers to day 5 or day 13. Please be clear about it.
· line 414: as well as other abiotic stresses such as heat, drought, freezing, etc.
· line 416: please replace "a" with "the"
· line 422: nitrate index, or nitrogen index?
· line 423: Please replace "meanwhile" with "however"
· End of Conclusions – please add the reservation about the expected potential impact of P. plecoglossicida on resistance against abiotic stress.

English language is mostly alright.
Author Response
General response: Dear reviewer, this is one of numerous articles we wrote, but we have never happened to get such an attentive reviewer like you. Our most sincere thanks for being so patient and friendly and providing so many valuable comments! We are sorry for having made so many mistakes, which you kindly corrected
- Manuscript title: please add a dash between "acid" and "bacteria" ("abscisic acid-degrading bacteria"). Please also make the same correction elsewhere in the manuscript, as appropriate.
Response: dash was added in the title and in two more places as well as checked everywhere.
- line 23: please correct "in soil and plants" into "in soil and plant shoots", as Figure 7 shows, that the increase in endogenous ABA was found only in shoots, but not in roots of the densely planted plants.
Response: Thanks for having noticed this. “Shoots” was added
- line 24: please re-organize the order of words in this sentence ("but accumulation of this hormone, as well as inhibition of the growth of grouped plants,was prevented by bacteria").
Response: The order of the words was reorganized according to the suggestion
- line 57: please replace the citation of Vysotskaya et al. 2011 with the reference number
Response: the name was substituted with the number [11]
- : please add a comma after "xylem sap"
Response: comma was put after “xylem sap”
- line 78: please provide the reference immediately after this sentence, even though it might be a repetition of a reference cited immediately previously or afterwards. Currently it is not obvious which reference this text refers to.
Response: Reference to [12] is introduced
- line 83: their ABA concentration in which tissue?And please provide the reference again immediately after this sentence.
Response: We added that ABA concentration was reduced in shoots and roots (as is said in figures 5 c and f of the article). Reference [24] is introduced.
- line 198: please put pin italic here too.
Response: p is in italic now. We are sorry for having missed this
- line 202: please add the article ("containing asingle plant")
Response: “a” is added
- line 206: please delete the word "occasion". Or you can replace it with "watering event" as you wrote afterwards. "Watering event" may sound appropriate, but "occasion" is not really a good word.
Response: "occasion" was substituted with "event"
- line 206: also please add a comma after "capacity"
Response: comma was added after “capacity”,
- line 210: I see another interesting thing in this graph, that might be worth a comment. The soil water content seemed to additionally decrease in the 108 treatment (compared to both non-inoculated plants, and 106-inoculated plants) in both individually and densely planted plants. How would you explain this? I see two possible explanations: (1) the higher bacterial inoculum might have stimulated the plants to extract the water from the soil more efficiently than they did in the absence (or presence of lower inoculum) of bacteria; or (2) it might be that the bacteria themselves extracted the water from the soil when inoculated at higher density. Although I'm not sure whether bacteria would have sufficient biomass to do that on a measurable scale. Or perhaps they do, if this strain forms a biofilm? In any case, I would like to read a commentary on this phenomenon, and I think it can be added directly to the Results section (not even to the Discussion), because it would serve more as a direct commentary on the displayed result, than as a discussion which is tightly related to your main research goal.
Response: Thanks for this interesting observation and provided explanation. I hope our respected reviewer wont’ be against our using exact phrases from the comments. We inserted that: “The soil water content seemed to additionally decrease in the 108 treatment (compared to both non-inoculated plants, and 106-inoculated plants) in both individually and densely planted plants. It is likely that the higher bacterial inoculum might have stimulated the plants to extract the water from the soil more efficiently than they did in the absence (or presence of lower inoculum) of bacteria”
- Figure legends (for all Figures): Since you decided to stick to the labels "106" and "108" throughout the manuscript text (instead of "3.8 x 106" and "3.8 x 108") I would prefer that you just write "106" and "108", because mentioning the measurement unit "CFU/g" suggests the exact density, which would be incorrect.
Response: According to this recommendation we deleted CFU/g everywhere including figure captures except the place, where we explain this designation (suspension of microorganisms diluted in sterile tap water was added to the pots to yield 3.8 x 106 and 3.8 x 108 CFU per g of soil thereafter designated as the 106 and 108 treatments, respectively).
- line 222: Here you use the British English spelling "neighbours", whereas I believe that in most of the rest of the manuscript you chose American English spelling for "neighbors", and for most other words. Please check the manuscript thoroughly for the consistent use of either British or American English.
Response: Now it is neighbors everywhere, except the title of one of articles in the list.
- line 233-234: You doubled the time designation ("by this time/at this time"). Please remove one of them.
Response: “at this time” was deleted. We are sorry for this doubled designation of the time
- line 237: Please, again delete the measurement unit "CFU/g"(or add the 3.8 x factor before the number).
Response: We have done this everywhere.
- line 256: Please add "of inoculum density" ("the effect of inoculum density")
Response: I hope we understood this remark properly. Now it is “the higher the inoculum density, the greater the effect”
- line 269, 275, 316: Why four days? I believe it should be five days.
Response: We counted this from the day of inoculation, which was one day after transplanting. But respected reviewer must be right and the date should be counted from the start of competition experiments. So it is five days now.
- line 269-271: It is necessary to emphasize, together with the sentence citing Figure 4A, that this effect was statistically significant in the non-inoculated plants, while in the inoculated plants it was observed too, but was not confirmed as statistically significant.
Response: Again we need to apologize for using the phrases proposed by respected reviewer. We added: “This effect was statistically significant in the non-inoculated plants, while in the inoculated plants it was observed too, but was not confirmed as statistically significant.
- line 274: Please add "leaf" ("between the leafweight of single and competing plants")
Response: “leaf” was added
- line 276-278: The statement citing Figure 4D is incomplete or inaccurate.You wanted to say that at day 13, it was recorded that the competition factor, in non-inoculated plants, increased the root-to-shoot ratio (an increase from 0.45 to 0.62 is not a 1.5-fold increase, please recalculate). You should also emphasize that this effect of competition on the root-to-shoot ratio, was decreased by inoculation, so the difference between single and competing plants was less pronounced in inoculated plants.
Response: Thanks for this comments and it is a shame that we described it so incompletely and inaccurately. I am afraid this resulted from the changes in the structure of the Results. This part is now written as follows: “However later on (on the 13th day), competition increased the ratio of the mass of roots and shoots by 1.4 times in non-inoculated plants (from 0.45 to 0.62 in single and grouped plants, respectively (Figure 4D)). This effect of competition on the root-to-shoot ratio, was decreased by inoculation, so the difference between single and competing plants was less pronounced in inoculated plants.”
- line 280-281: In the sentence citing Figure 5A, it should be added that this relationship, however, was confirmed by Duncan's test as statistically significant only at the 106treatment.
Response: Yes, it is so and we added that “This relationship, however, was confirmed by Duncan's test as statistically significant only at the 106 treatment”
- Figure 7: Please add directly to the figure (either on the y-axis or as a title above each subfigure) the designations "shoots", "roots", "soil" so that the reader does not have to scroll down to read the figure caption to remember what each graph is about.
Response: "shoots", "roots", "soil" is added to y-axis of corresponding figures
- Discussion– Discussion is really greatly improved and I want to congratulate you on the new version of the Discussion, which I like very much. I also welcome the fragmentation of the Discussion section into subsections, however I find that the subsection titles are too long and complicated, please shorten them down to make them easier to grasp
Response: Thanks for your kind words. We shortened two really long subtitles: “4.2.1.Bacterial effects on concentration of abscisic acid in culture media, soil and plants.” And “4.2.2. Synthesis of abscisic acid by plants and its bacterial degradation
- line 321 – "detected in our experiments at the beginning of the competition" – this statement is inaccurate.The second part of this same statement (correctly) states that there was an absence of change in this indicator at the beginning of the experiment.
Response: We are sorry for this silly sentence. It is now modified and goes as follows: “An increase in the mass ratio of roots and shoots (relative activation of root 320 growth) is a classic response to water deficiency [43,44] and the absence of changes in this indicator is in accordance with the absence of water shortage detected at the beginning of the competition experiment.”
- line 328-329 – this sentence too is not completely precise, even though it is not really incorrect. The plants indeed did not suffer from nitrogen deficiency, but the soil water content was even lower in inoculated than in non-inoculated plants – which might mean that the water uptake was more efficient in inoculated plants, which may be the cause why they did not have to increase their investment in root growth compared to single plants. Please make these statements more detailed and clear.
Response: We tried to clarify this by adding that “water uptake was more efficient in inoculated plants compared to single plants without investing in root growth.”
- line 350-351: At this point, the readers might be interested to get to know briefly about how Rhodococcus(by which mechanism) was able to reduce the endogenous levels of ABA, i.e., in which plant tissues.
Response: To address this comment I reread the article of Belimov et al. and failed to find any explanation there. It is only stated that this bacteria were able to degrade ABA and that ABA concentration was decreased by inoculation with this bacteria. Authors even did not insist on direct causal relationship between the effects,
- line 362-364: This sentence is too complicated and very difficult to follow, please make it easier to read.
Response: To clarify importance of the ability of bacteria to degrade ABA for the control of the growth of grouped plants, we added that “ABA concentration increased in uninoculated grouped plants, which correlated with inhibition of plant growth due to competition, whereas inoculation of plants with ABA-degrading bacteria led to a decrease in ABA concentration in grouped plants, which was accompanied by a decrease in growth inhibitory effect of competition.” We hope that this sentence justifies further conclusion that “These results confirm importance of the ability of bacteria to degrade ABA for the control of the growth of grouped plants that had increased concentration of ABA when not treated with bacteria”
- line 371-373: It is unclear whether this sentence refers to day 5 or day 13. Please be clear about it.
Response: In accordance we added that the end of experiment is meant, when grouped plants extracted more water than single plants.
- Line 414: as well as other abiotic stresses such as heat, drought, freezing, etc.
Response: We added just this phrase after “salinity” (“as well as other abiotic stresses such as heat, drought, freezing and others”)
- line 416: please replace "a" with "the"
Response: “a” was replaced with “the”
- line 422: nitrate index, or nitrogen index?
Response: Sorry! It should be nitrogen index. This was corrected.
- line 423: Please replace "meanwhile" with "however"
Response: "meanwhile" was replaced with "however"
- End of Conclusions – please add the reservation about the expected potential impact of plecoglossicidaon resistance against abiotic stress.
Response: In accordance with this recommendation we added that “However, application of this approach will depend on further study of the effect of inoculation with ABA-degrading bacteria on plant resistance against abiotic stress.”
Reviewer 3 Report
Comments and Suggestions for Authors
The authors are improving the paper according to my comments. Only minor editing of English language required
Comments on the Quality of English LanguageOnly minor editing of English language required
Author Response
We are happy that we managed to satisfy the reviewer with our revision (thanks once again for the valuable comments). We went through the text and corrected minor mistakes in English mentioned by the reviewer